# The C-type lectin domain of CD62P (P-selectin) functions as an integrin ligand

Yoko K Takada[1], Scott I Simon[1,2], Yoshikazu Takada[1,3]

**Recognition of integrins by CD62P has not been reported and this motivated a docking simulation using integrin $\alpha v\beta3$ as a target. We predicted that the C-type lectin domain of CD62P functions as a potential integrin ligand and observed that it specifically bound to soluble $\beta3$ and $\beta1$ integrins. Known inhibitors of the interaction between CD62P–PSGL-1 did not suppress the binding, whereas the disintegrin domain of ADAM-15, a known integrin ligand, suppressed recognition by the lectin domain. Furthermore, an R16E/K17E mutation in the predicted integrin-binding interface located outside of the glycan-binding site within the lectin domain, strongly inhibited CD62P binding to integrins. In contrast, the E88D mutation that strongly disrupts glycan binding only slightly affected CD62P-integrin recognition, indicating that the glycan and integrin-binding sites are distinct. Notably, the lectin domain allosterically activated integrins by binding to the allosteric site 2. We conclude that CD62P-integrin binding may function to promote a diverse set of cell–cell adhesive interactions given that $\beta3$ and $\beta1$ integrins are more widely expressed than PSGL-1 that is limited to leukocytes.**

## Introduction

CD62P (P-selectin), a member of the selectin family, has been identified as a $Ca^{2+}$-dependent receptor that binds to carbohydrates on neutrophils and monocytes. CD62P is stored in the $\alpha$-granules of platelets and Weibel–Palade bodies of endothelial cells and is transferred to the membrane upon activation (Palabrica et al, 1992; Mayadas et al, 1993; Wagner, 1993). All selectins are composed of three distinct domains: a C-type lectin-like domain in the N-terminus, an EGF-like domain, and a complement-binding protein-like domain composed of short consensus repeats (~60 amino acids). CD62P is anchored in the plasma membrane by a short cytoplasmic tail (Vestweber and Blanks, 1999). The lectin domain of CD62P recognizes sialyl-Lewis X on its glycoprotein ligand-1 (PSGL-1) to mediate rapid rolling of leukocytes over vascular surfaces during the initial steps in inflammation (Springer, 1994). Thus, CD62P is a major therapeutic target for cardiovascular disease, inflammation, and in cancer metastasis (Ludwig et al, 2007).

Integrins are a family of cell-surface $\alpha\beta$ receptor heterodimers that bind to extracellular matrix ligands (e.g., fibronectin, fibrinogen, and collagen), cell-surface ligands (e.g., ICAM-1 and VCAM-1), and soluble ligands (e.g., growth factors) (Takada et al, 2007). By virtual screening of the protein data bank (PDB) employing the integrin headpiece as a target for docking simulations, we discovered new potential integrin ligands. These ligands have been shown to bind to the classical ligand-binding site of integrins (site 1) defined by the crystal structure of the RGD–$\alpha v\beta3$ complex (Xiong et al, 2001, 2002). The top face of the $\beta$I domain contains three cation-binding sites: the metal ion-dependent adhesion site (MIDAS), the ADMIDAS (adjacent to MIDAS), and LIMBS (ligand-associated metal-binding site) (Valdramidou et al, 2008; Anderson et al, 2022). Furthermore, the $\beta$I domain contains the loop sequence that controls ligand binding specificity (i.e., 177CYDMKTTC184, specificity-determining loop, SDL) (Takagi et al, 1997; Artoni et al, 2004).

We have discovered several novel ligands that recognize the allosteric binding site of integrins (site 2) and allosterically activate them in the absence of canonical inside-out signaling. These ligands include CX3CL1 (Fujita et al, 2014), CXCL12 (Fujita et al, 2018), CCL5 (Takada et al, 2022), CD40L (Takada et al, 2021), and sPLA2-IIA (Fujita et al, 2015). The position of site 2 on the opposite side of site 1 in the integrin headpiece was identified by docking simulations of the interaction between inactive (closed headpiece) integrin $\alpha v\beta3$ and CX3CL1 (Fujita et al, 2014). Moreover, peptides from site 2 bound to these ligands and suppressed integrin activation, indicating that the binding of these ligands to site 2 induce allosteric activation in the absence of global conformational changes (Fujita et al, 2014, 2015). In addition, it has been reported that 25-hydroxycholesterol (25HC), a pro-inflammatory cholesterol metabolite, bound to site 2 of integrins and induced integrin activation and inflammatory signaling (Pokharel et al, 2019). Taken together, these data implicate site 2 in recognition of a diverse set of ligands that can induce inflammatory signaling.

[1]Department of Dermatology, UC Davis School of Medicine, Sacramento, CA, USA [2]Department of Biomedical Engineering, UC Davis, Davis, CA, USA [3]Department of Biochemistry and Molecular Medicine, UC Davis School of Medicine, Sacramento, CA, USA

Correspondence: ytakada@ucdavis.edu

Here, we report that the CD62P lectin domain functions as a potential integrin ligand based upon virtual screening of the PDB. Because a literature search did not locate a report describing integrin interaction with CD62P, we performed a CD62P-$\alpha v \beta 3$ docking model and discovered a potential binding site on CD62P distinct from that of a glycan-binding site that recognizes the classical site 1 on integrins. We identified amino acid residues critical for integrin binding to the CD62P lectin domain by introducing mutations in the predicted integrin-binding site (e.g., the R16E/K17E mutation). The E88D mutation that is known to block glycan binding (Mehta-D'souza et al, 2017) was found to minimally affect integrin binding. Therefore, we propose that CD62P functions as an integrin ligand after its membrane up-regulation on activated endothelial cells or platelets and could support cell–cell adhesion by binding to integrins, in addition to mediating glycan binding and rolling. In addition, the CD62P lectin domain bound to site 2 and activated recombinant soluble integrins and those expressed on cell membranes.

# Results

## The CD62P (P-selectin) lectin domain specifically binds to soluble integrins $\alpha v \beta 3$ and $\alpha IIb \beta 3$

We virtually screened the PDB for potential integrin ligands employing a docking simulation with integrin $\alpha v \beta 3$ (1L5G.pdb, open headpiece) as a target. The simulation predicted that the CD62P lectin domain could function as a potential integrin ligand. This prediction is independent of current models of CD62P, which recognizes sialyl-Lewis X expressed on PSGL-1 that mediates rapid rolling of leukocyte over vascular surfaces during the initial steps in inflammation.

To test this prediction, we assessed if the CD62P lectin domain (residues 1–117) and the combined lectin and EGF-like domain (residues 1–158) (Fig 1A) binds to recombinant integrins. We found that soluble $\alpha v \beta 3$ and $\alpha IIb \beta 3$ bound in a dose-dependent manner to the CD62P lectin domain in ELISA-based assay following integrin activation in the presence of $Mn^{2+}$ (TH-1 mM $Mn^{2+}$) (Fig 1B). The CD62P lectin domain exhibited twofold stronger binding to integrins than the combined lectin and EGF-like domains (Fig 1C), indicating that the CD62P lectin domain is primarily involved in integrin recognition.

For the remainder of these studies, we focused on examining the interaction between the CD62P lectin domain as a ligand for $\alpha IIb \beta 3$ and $\alpha v \beta 3$. Because RGD peptides or $7 \times 10^3$ (anti-$\beta 3$) did not affect the binding of the lectin domain to soluble $\alpha v \beta 3$ (Fig S1B), we examined additional known ligands for $\alpha IIb \beta 3$ and $\alpha v \beta 3$ to compete for binding to the lectin domain. The ADAM15 disintegrin domain has been reported to be a specific ligand for $\alpha v \beta 3$ (Zhang et al, 1998) and $\alpha IIb \beta 3$ (Langer et al, 2005). We observed that ADAM15 disintegrin fused to GST suppressed the binding of soluble $\alpha v \beta 3$ or $\alpha IIb \beta 3$ to the immobilized lectin domain, but parent GST did not, indicating that the CD62P lectin domain competes with ADAM15 for binding to integrins (Fig 1D).

Heat-inactivation reduced integrin binding, indicating that the lectin domain requires proper folding for integrin function (Fig 1E

and F). Also, CD62P lectin domain exhibited cation-dependence for binding to soluble integrins $\alpha v \beta 3$ and $\alpha IIb \beta 3$ (1 mM $Mn^{2+}$>$Mg^{2+}$> $Ca^{2+}$>EDTA) (Fig 1E and F) with a hierarchy that is similar to that of known integrin ligands. These findings are consistent with the observation that the CD62P lectin domain functions as a specific ligand for $\alpha v \beta 3$ and $\alpha IIb \beta 3$.

## The integrin-binding site and the glycan-binding site are distinct

The structure of PSGL-1 peptide (605YEYLDYDFLPETEP618) in the PSGL-1-CD62P lectin domain complex has been resolved (1G1S.pdb) (Somers et al, 2000). It has been proposed that ligand binding to this lectin domain closes loop 83–89 around the $Ca^{2+}$ coordination site, enabling Glu-88 to engage $Ca^{2+}$ and fucose (Somers et al, 2000). All three selectins require Glu-88 to sustain bonds with $sLe^x$-containing ligands under repulsive forces. Furthermore, it is reported that mutating Glu-88 to Asp (the E88D mutation) locks selectins in their functionally inactive states and markedly impairs selectin-mediated rolling of transfected mouse L-1 B cells under flow (Mehta-D'souza et al, 2017). We generated a model of the possible interactions between the integrin, CD62P lectin domain, and PSGL-1 by superposing the CD62P lectin domain-integrin $\alpha v \beta 3$ docking model and the PSGL-1–CD62P lectin domain complex. The model predicts that CD62P lectin domain can bind to integrin $\alpha v \beta 3$ and PSGL-1 in the absence of steric hindrance (Fig 2A). The specificity loop (residues 177–184 of $\beta 3$) (Takagi et al, 1997) (Fig 2B) located next to site 1 (see Fig 4A) is shown in the docking model to clarify the position of site 1. To determine if the glycan and integrin-binding sites are distinct, several amino acid residues in the predicted integrin-binding interface of the lectin domain were selected for mutagenesis to Glu (Fig 2B). These amino acid residues are conserved among selectins as shown in Fig 2C. Several mutants (R16E/K17E, K58E, K66E/K67E, K84E/R85E) were defective in binding to soluble $\alpha IIb \beta 3$ and $\alpha v \beta 3$ activated in the presence of 1 mM $Mn^{2+}$ (Fig 2D and E). Notably, the E88D mutation did not affect integrin binding, indicating that the glycan and integrin-binding sites are distinct. However, because the K84E/R85E mutation reduced integrin binding, it is likely that the glycan and integrin-binding sites may be close to, or overlap, each other as indicated by the amino acid residues in Fig 2C. The R54E/K55E mutant showed higher binding to integrins than WT lectin domain (gain of function mutant).

## Inhibitors of CD62P–PSGL-1 interaction did not block the lectin domain–integrin interaction

Because the binding sites for glycan ligands and integrins are close to each other in the lectin domain, it is possible that currently available antagonists to CD62P could inhibit integrin binding. However, competitive addition of a PSGL-1-Fc fusion protein did not affect the binding of soluble integrins $\alpha v \beta 3$ and $\alpha IIb \beta 3$ to the lectin domain (Fig S1C). We found that a widely used monoclonal antibody P8G6 against CD62P did not reduce integrin binding to the lectin domain (Fig S1D). This antibody has been reported to block CD62P-induced platelet aggregation (Theoret et al, 2006). We also tested a non-carbohydrate small-molecular weight CD62P inhibitor KF38789 that blocks adhesion of U937 monocytic cells, which express both

## a

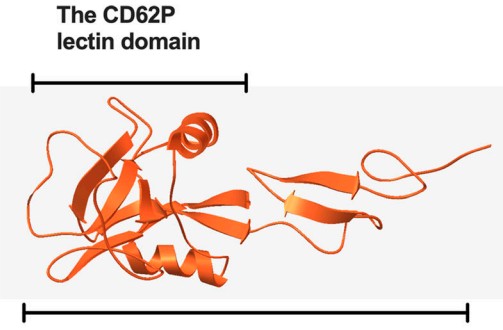

The CD62P lectin domain

The CD62P lectin plus EGF-like domains

## b

Soluble integrins bind to the of CD62P lectin domain in 1 mM $Mn^{2+}$

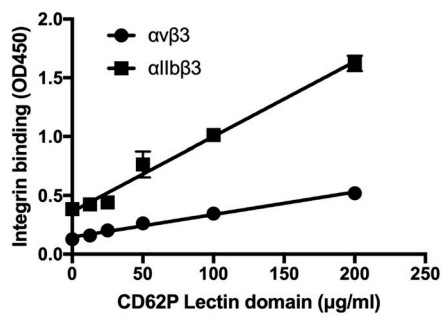

## c   The CD62P lectin domain is primarily involved in integrin binding

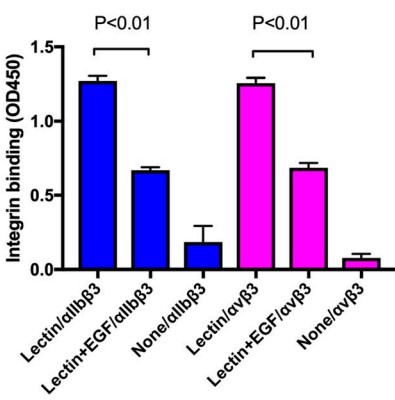

## d   ADAM15 suppresses integrin binding to the CD62P lectin domain

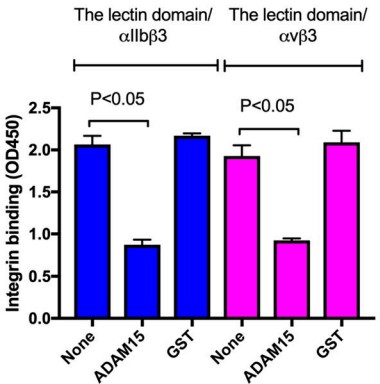

## e   The CD62P lectin domain- integrin $\alpha v\beta 3$ interaction is cation-dependent

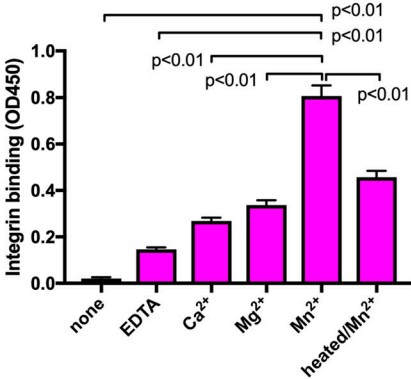

## f   The CD62P lectin domain-integrin $\alpha IIb\beta 3$ interaction is cation-dependent

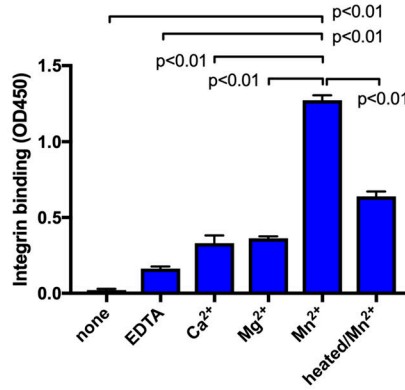

**Figure 1. The lectin domain bound to soluble integrins $\alpha v\beta 3$ or $\alpha IIb\beta 3$ in ELISA-type binding assays.**
**(A)** The CD62P lectin and the EGF domains in the crystal structure of CD62P. **(B)** Binding of soluble integrins to the immobilized lectin domain of CD62P. Wells of the 96-well microtiter plate were coated with the CD62P lectin domain and the remaining protein-binding sites were blocked with BSA. Wells were incubated with soluble integrin $\alpha v\beta 3$ or $\alpha IIb\beta 3$ (1 μg/ml) for 1 h in 1 mM $Mn^{2+}$ and bound integrins were quantified using anti-$\beta 3$ mAb and anti-mouse IgG conjugated with HRP. **(C)** The CD62P lectin domain binds better to soluble integrins than the combined lectin and EGF domains (at a coating conc. of 50 μg/ml). **(D)** The disintegrin domain of ADAM15, another ligand for $\alpha v\beta 3$ or $\alpha IIb\beta 3$ suppresses the binding of soluble integrins to the lectin domain. **(E, F)** The binding of soluble integrins to the immobilized CD62P lectin domain (coating concentration at 50 μg/ml) in 1 mM different cations. Data are shown as means ± SD (n = 3). Statistical analysis was performed by ANOVA in Prism 7.

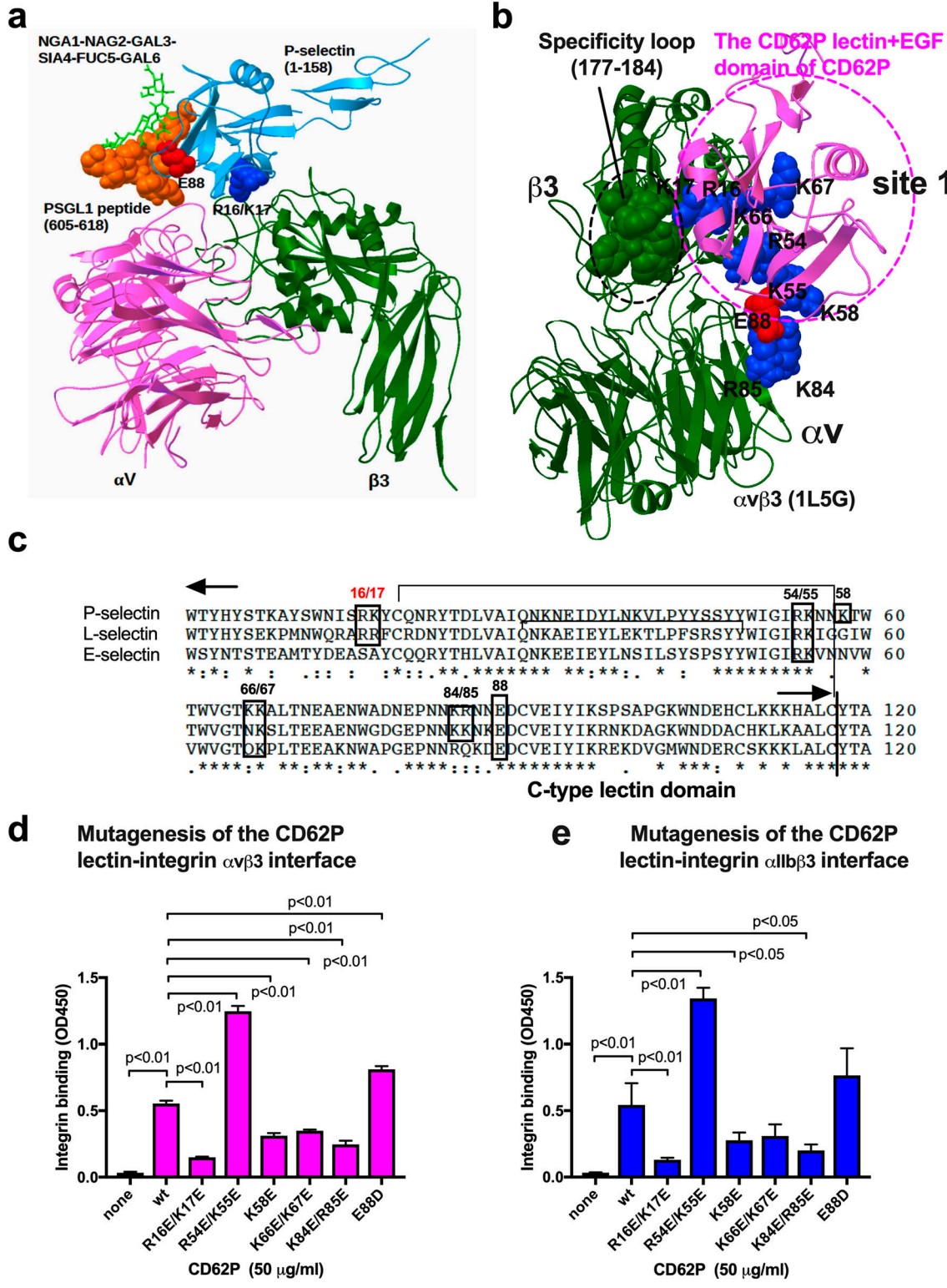

**Figure 2. The CD62P lectin domain binding to integrin and PSGL-1.**
**(A)** The CD62P-αvβ3 docking model was superposed with the crystal structure of the CD62P–PSGL1 peptide complex (1G1S.pdb). The superposed model predicts that integrin-binding site and PSGL-1-binding site are distinct. R16/K17 of the CD62P lectin domain is close to integrin αvβ3 and E88 of the lectin domain is close to PSGL-1 peptide and glycan. **(B)** Docking simulation of the interaction between open/active αvβ3 (1L5G.pdb) and the CD62P lectin domain (1G1Q.pdb) was performed using Autodock3. The specificity loop (residues 177–184 of β3) is next to site 1. This loop is located between site 1 and site 2 (Fig 4A). The amino acid residues selected for mutagenesis are shown. **(C)** Alignment of P-, L-, and E-selectins. R16/K17 and E88 are conserved in selectins. **(D, E)** The binding of the CD62P lectin domain mutants to soluble integrins αvβ3 or αIIbβ3. Data are shown as means ± SD (n = 3). Statistical analysis was done by ANOVA in Prism7.

PSGL-1 and integrins, to immobilized CD62P (Ohta et al, 2001). This inhibitor did not block integrin binding to the lectin domain (Fig S1E). These findings are consistent with a docking model in which integrins and glycans simultaneously bind to distinct sites on the lectin domain.

### The lectin domain supports static cell adhesion in a PSGL-1-independent manner

It is known that CHO cells lack two enzymes, the core 2 $\beta$1-6-N-acetylglucosaminyltransferase and an $\alpha$1-3 fucosyltransferase, which mediate branching and fucosylation of the O-linked glycans on PSGL-1. Therefore, CHO cells can synthesize PSGL-1 core protein by transfection of its cDNA, but PSGL-1 core protein cannot bind to CD62P in the absence of glycosylation (Li et al, 1996). CHO cells were used to test the binding of the CD62P lectin domain to integrins in a PSGL-1-independent manner. We found that immobilized WT CD62P lectin domain supported adhesion of CHO cells that express recombinant $\alpha$v$\beta$3 ($\beta$3-CHO cells) and parent CHO cells (Fig 3A and B). Integrins are not fully activated in DMEM due to high [$Ca^{2+}$] (>1 mM), whereas the WT lectin domain supported cell adhesion more strongly in Tyrode-HEPES buffer with 1 mM $Mg^{2+}$ (Kd = 15.8–19.9 $\mu$g/ml), in which integrins are more activated than in DMEM (Kd could not be measured). The CD62P lectin domain supported the adhesion of CHO and $\beta$3-CHO cells in 1 mM $Mg^{2+}$ more strongly than control BSA in a time-dependent manner (Fig 3C and D). This is consistent with the idea that the CD62P lectin domain supports cell adhesion specifically by binding to activated integrins on an adjacent cell. E88D supported integrin-mediated cell adhesion to a level comparable with that of WT CD62P in 1 mM $Mg^{2+}$. This is quite different from the effect of the E88D mutation on rolling of transfected mouse L-1 B cells on PSGL-1 under flow (Mehta-D'souza et al, 2017), indicating that the CD62P lectin domain–integrin interaction is distinct from its recognition of PSGL-1. The R16E/K17E mutation, which reduced integrin binding in ELISA-type binding assay, showed 60% reduced cell adhesion in 2 mM $Mg^{2+}$ and did not support cell adhesion in DMEM. Furthermore, Arg-16 and Lys-17 are not part of the glycan-binding region of CD62P, which is consistent with the idea that glycan binding and integrin binding are separate functions of CD62P (Fig 3E and F). Taken together, these data indicate that the CD62P lectin domain supports adhesion of CHO cells in a cation-dependent and PSGL-1-independent manner.

### The lectin domain of CD62P activates soluble integrins $\alpha$v$\beta$3 and $\alpha$IIb$\beta$3 in 1 mM $Ca^{2+}$ in cell-free conditions

It has been proposed that CD62P primes leukocyte integrin activation during rolling to arrest on inflamed endothelium (Wang et al, 2007). However, the specifics of the mechanism of priming have not been established. Integrins exist in an inactive state in the presence of high [$Ca^{2+}$] (in DMEM) and require activation to interact with ligands. It has been proposed that platelet integrin $\alpha$IIb$\beta$3 is activated by inside-out signaling that is triggered by thrombin or other platelet agonists binding to cell-surface receptors (Han et al, 2006; Shattil et al, 2010; Ginsberg, 2014). We have reported that several integrin ligands (e.g., fractalkine, SDF-1, sPLA2-IIA, and CD40L) bound directly to the allosteric site of integrins (site 2) and

activated soluble integrins in cell-free conditions (Fujita et al, 2014, 2015, 2018; Takada et al, 2021). Docking simulation of the interaction between the lectin domain and $\alpha$v$\beta$3 (1JV2.pdb, closed headpiece) predicted that the lectin domain binds to site 2 relatively tightly (i.e., −20.18 kcal/mol) (Fig 4A). The specificity loop (residues 177–184 of $\beta$3) (Takagi et al, 1997), which is located between site 1 and site 2 is shown to clarify the positions of these site 2. To assess the function of CD62P lectin domain in allosterically activating soluble $\alpha$v$\beta$3 and $\alpha$IIb$\beta$3, an ELISA-based activation assay was employed. Wells of 96-well microtiter wells were coated with fibrinogen fragments, $\gamma$C399tr specific to $\alpha$v$\beta$3 or $\gamma$C390-411 specific to $\alpha$IIb$\beta$3, and subsequently incubated in wells with soluble integrins $\alpha$v$\beta$3 and $\alpha$IIb$\beta$3 in the presence of the lectin domain in 1 mM $Ca^{2+}$ (to keep integrins inactive). Activated soluble integrins that bound to fibrinogen fragments immobilized to the well bottom were quantified by secondary labeling with anti-$\beta$3 antibody. We found that the CD62P lectin domain enhanced the binding of both soluble $\alpha$v$\beta$3 and $\alpha$IIb$\beta$3 integrins to their cognate ligands in a dose-dependent manner (Fig 4B and C). This indicates that the lectin domain activated soluble $\alpha$v$\beta$3 and $\alpha$IIb$\beta$3 in cell-free conditions in an allosteric manner.

Because amino acid residues of the CD62P lectin domain that are involved in site 2 binding overlap with those involved in site 1 binding (Fig 4A and Tables 1 and 2), we next assessed if mutations of the CD62P lectin domain affect integrin activation upon binding to $\alpha$v$\beta$3 or $\alpha$IIb$\beta$3. Notably, most of the mutations tested (K8E, R16E/K17E, R22E, R54E/K55E, K58E, K66R/K67E, and K84E/R85E) enhanced integrin activation by the CD62P lectin domain relative to the WT lectin domain, indicating that they are gain-of-function mutations. Remarkably, the pattern of increased binding was identical between $\alpha$v$\beta$3 and $\alpha$IIb$\beta$3 for each mutation (Fig 4D and E).

We next evaluated if the CD62P lectin domain activates cell-surface integrins $\alpha$v$\beta$3 and $\alpha$IIb$\beta$3 using $\beta$3-CHO cells or CHO cells that express $\alpha$IIb$\beta$3 ($\alpha$IIb$\beta$3-CHO). A cell-based flow cytometry assay was used to quantify the binding of FITC-labeled fibrinogen fragments $\gamma$C399tr or $\gamma$C390-411 to cell-expressing integrins in the presence of 1 mM $Ca^{2+}$ (to maintain integrin inactivity) in the presence or absence of the lectin domain. We found that ligand binding to cells was markedly enhanced in the presence of the CD62P lectin domain, indicating that $\alpha$v$\beta$3 and $\alpha$IIb$\beta$3 on the cell surface were activated (Fig 5A–D).

### The CD62P lectin domain binds to peptides from site 2

We previously published that amino acid residues from site 2 are predicted to be close to site 2 ligands (Table 2). We found that linear peptides from site 2 (residues 267–287 of $\beta$3 and residues 275–294 of $\beta$1) bound to several site 2 ligands (e.g., CX3CL1, CXCL12, CCL5, sPLA2-IIA, and CD40L) (Fujita et al, 2014, 2015, 2018; Takada et al, 2021), strongly indicating that they bind to site 2. We predict that the lectin domain of CD62P binds to site 2 and activates integrins based on the docking simulation prediction. However, it is unclear if this is the case in the lectin domain. We generated cyclic site 2 peptides (residues 260–288 of $\beta$3 and residues 268–295 of $\beta$1) to stabilize site 2 peptides (Takada et al, 2022). In the present study, the $\alpha$v$\beta$3-CD62P lectin domain docking model predicts that site 2 interacts

**a** **Dose-dependent adhesion of CHO cells to the CD62P lectin domain**

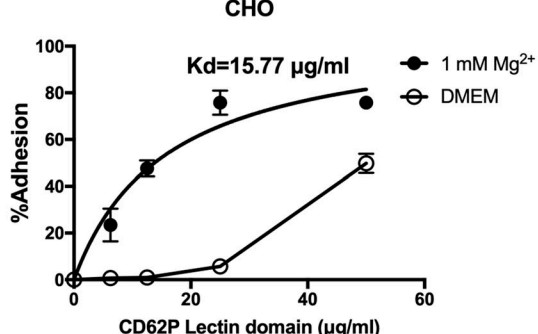

**b** **Dose-dependent adhesion of β3-CHO cells to the CD62P lectin domain**

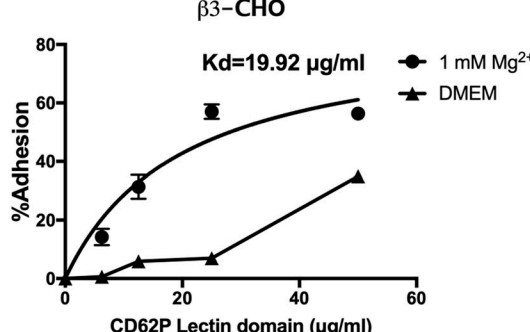

**c** **Time-course of adhesion of CHO cells to the CD62P lectin domain**

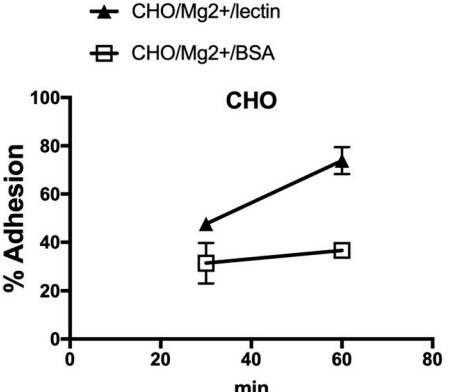

**d** **Time-course of adhesion of β3-CHO cells to the CD62P lectin domain**

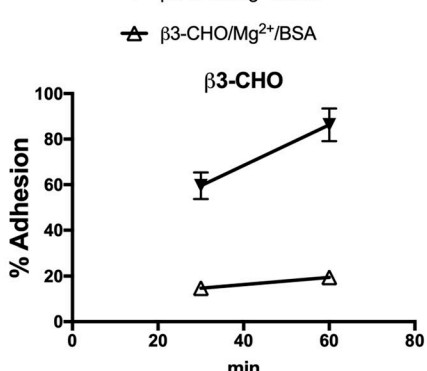

**e** **Effect of mutations on adhesion of CHO cells to the CD62P lectin domain**

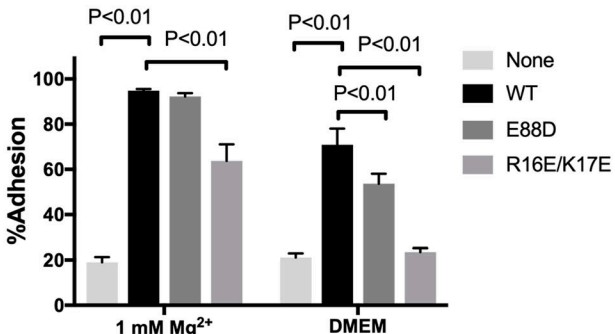

**f** **Effect of mutations on adhesion of β3-CHO cells to the CD62P lectin domain**

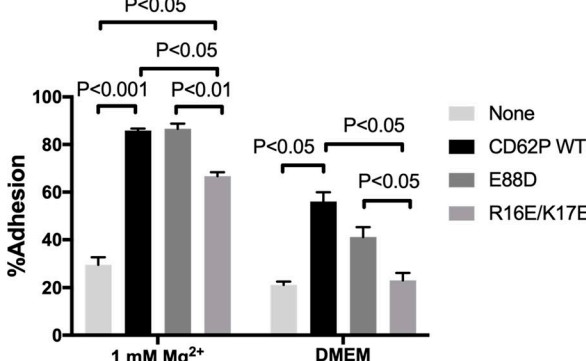

**Figure 3. Adhesion of CHO cells (PSGL-1 negative) to the CD62P lectin domain.**
PSGL-1 is expressed in leukocytes, but not in CHO cells. **(A, B)** Effect of cations on cell adhesion to the CD62P lectin domain. The wells were coated with WT CD62P lectin domain and the remaining protein-binding sites were blocked with BSA. The wells were then incubated with CHO or β3-CHO cells for 1 h at 37°C in DMEM or Tyrode–HEPES buffer with 1 mM Mg²⁺ and bound cells were quantified after brief rinsing using endogenous phosphatase activity. **(C, D)** Time-course of cell adhesion to the CD62P lectin domain. Wells were coated with the lectin domain (50 μg/ml) and remaining protein binding sites were blocked with BSA. The wells were incubated with CHO or β3-CHO cells for 30 or 60 min in 1 mM Mg²⁺ and bound cells were quantified as described above. **(E, F)** Effect of the lectin domain mutants on adhesion to the CD62P lectin domain. Wells of 96-well microtiter plate were coated with the CD62P lectin domain (WT or mutants, coating concentration at 50 μg/ml) and the remaining protein-binding sites

with the lectin domain (Fig 5E and Table 2). We found that cyclic site 2 peptides from β3 and β1 bound to the CD62P lectin domain in ELISA more strongly than control peptides (Fig 5F), indicating that the CD62P lectin domain binds to site 2.

### The CD62P lectin domain binds to and activates α5β1 and α4β1 integrins

Because parent CHO cells (α5β1+, αvβ3−) and those transfected to express αvβ3 adhered to the lectin domain at comparable levels, we next assessed the ability of CD62P lectin domain to bind to and activate the β1 integrins. Wells were coated with the CD62P lectin domain and bound biotinylated integrins were quantified using HRP-conjugated streptavidin. We observed that the CD62P lectin domain interacted with soluble α5β1 and α4β1 activated by 1 mM $Mn^{2+}$ (Fig 6A and D). To confirm the specificity of lectin binding of α4β1, we added either BIO1211, a specific antagonist to α4β1, or a fibronectin fragment H120 that serves as a specific ligand to α4β1; neither inhibiting lectin domain binding to α4β1 (Fig S1F and G). Because site 2 peptides from β1 bound to the CD62P lectin domain (Fig 6E), we expected that CD62P lectin domain binds to site 2 of β1 integrins and allosterically activate them. Flow cytometric analysis of the activation of cell-surface α5β1 on CHO and α4β1 on α4-CHO cells (which express recombinant human α4 on CHO cells) (Kamata et al, 1995) was detected by the binding of FITC-labeled fibronectin fragments in the presence or absence of the CD62P lectin domain (Fig 6B, C, E, and F). The CD62P lectin domain markedly enhanced the binding of FITC-labeled ligands, indicating that CD62P lectin domain can activate α5β1 and α4β1. These data indicate that the CD62P lectin domain-induced activation of these integrins may play a role in immune-competent cell binding to activated platelets or endothelial cells.

## Discussion

The present study establishes for the first time that the CD62P lectin domain specifically recognizes both β1- and β3-integrins and upon binding, allosterically activates cell adhesion to their respective cognate ligands on fibrinogen and fibronectin. Computational simulations using structural data from the PDB supported the premise that the CD62P lectin domain can dock to the headpiece of integrin αvβ3. We demonstrated that isolated CD62P lectin domain bound to soluble integrins αIIbβ3 and αvβ3 in a dose-dependent manner. The lectin domain proved to be a better ligand than the combined lectin and EGF-like domain in integrin binding. The CD62P lectin domain required cations for binding to integrins with the predicted hierarchy ($Mn^{2+} > Mg^{2+} > Ca^{2+} > EDTA$), which is like other known integrin ligands and indicates that an allosteric shift occurs to reveal the binding site. This finding extends the current concept that CD62P expressed on platelets and endothelium binds solely to glycan-presenting ligands and predicts that direct binding of the

CD62P lectin domain to integrins may be critically involved in CD62P-mediated cell–cell adhesion and signaling.

It is likely that the antagonists to integrins were selected for targets on known integrin ligands such as RGD-expressing fibronectin that are distinct from the CD62P lectin domain. We suspect that the binding site for the CD62P lectin domain and those of known inhibitors for RGD-based ligands do not overlap. One exception is the ADAM-15 disintegrin domain that suppressed CD62P lectin domain integrin binding. This indicates that the binding sites for CD62P lectin domain and the disintegrin domain overlap, suggesting that ADAM-15 may be able to block CD62P-binding to integrins in vivo. Also notable is that the binding of the CD62P lectin domain was not inhibited by known antagonists for glycan–CD62P interactions (KF38789, P8G6) (Fig S1). Moreover, soluble PSGL-1-Fc did not block integrin–CD62P interactions, suggesting that CD62P recognition of PSGL-1 and integrins may occur simultaneously. Thus, leukocyte rolling on endothelium expressing P-selectin may elicit activation of integrins from the outside–in to support signaling necessary for subsequent cell arrest and migration.

### Predictions of lectin–integrin interaction site

Docking simulation of the interaction between CD62P lectin domain and integrin αvβ3 predicts that PSGL-1-binding site and integrin-binding site on the lectin domain are close but distinct. We selected several amino acid residues in the predicted integrin-binding interface of the CD62P lectin domain for mutagenesis studies. Several mutations (particularly R16E/K17E) effectively reduced integrin binding, indicating that the structural model was accurate in predicting the docking site. In contrast, the E88D mutant of the CD62P lectin domain, which is known to affect glycan binding, did not affect integrin binding. Interestingly, the R54E/K55E mutant showed enhanced capacity to bind integrins, which may represent a gain-of-function mutation. If the CD62P lectin domain and its interactions with integrins and PSGL-1 can coexist on the cell surface, this implies that CD62P mutants defective in integrin binding (e.g., R16E/K17E) or glycan binding (e.g., D88E) may act as antagonists of signaling. It would be useful to develop antagonists that effectively block the CD62P lectin domain–integrin interaction for elucidating the physiological functions of the interaction of CD62P with integrins. Such a molecule could be used to delve the importance of CD62P lectin-dependent integrin signaling of leukocyte adhesion in inflammation, thrombosis, and metastasis, and perhaps for development of potential therapeutics.

We demonstrated that both soluble and membrane-expressed integrins bound to the immobilized CD62P lectin domain in relatively static adhesion assays. CHO cells lack the canonical sialylated PSGL-1 that supports CD62P-dependent cell tethering and rolling under shear stress (Li et al, 1996). A novel finding here is that the CD62P lectin domain supported static adhesion of CHO cells expressing recombinant αvβ3 and parent CHO cells in a cation-dependent manner (1 mM $Mg^{2+}$ > DMEM with high $[Ca^{2+}]$).

were blocked with BSA. Wells were incubated with CHO cells or β3-CHO cells in Tyrode–HEPES/1 mM $Mg^{2+}$ or DMEM. The E88D mutant is defective in binding to the glycan ligand and the R16E/K17E mutant is defective in integrin binding. Data are shown as means ± SD (n = 3). Statistical analysis was done by ANOVA in Prism7.

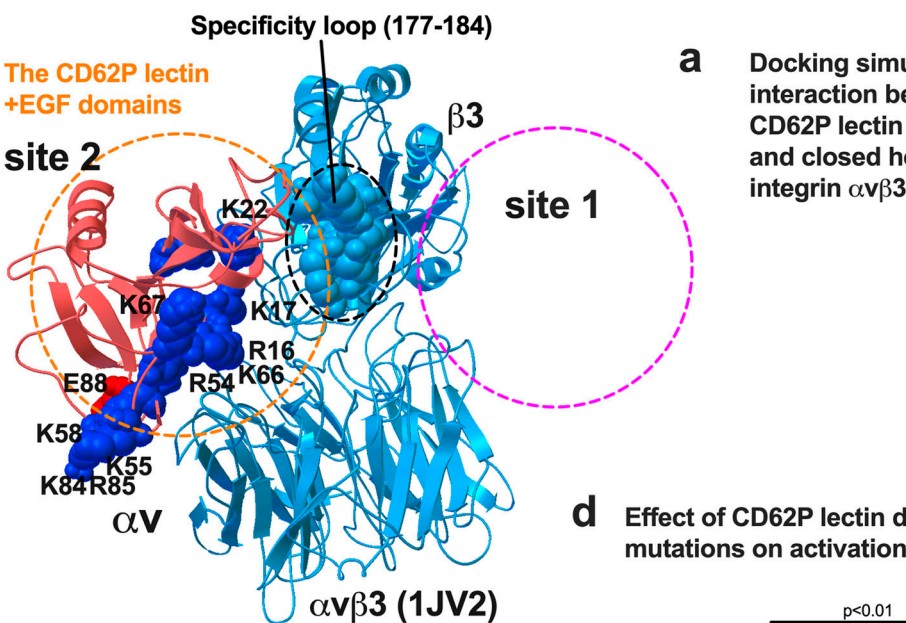

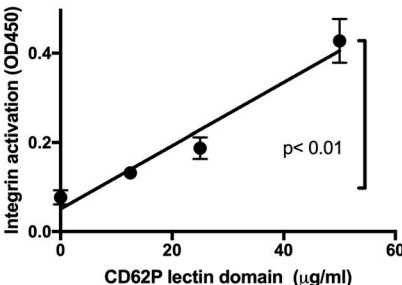

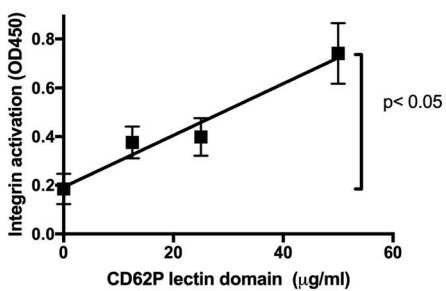

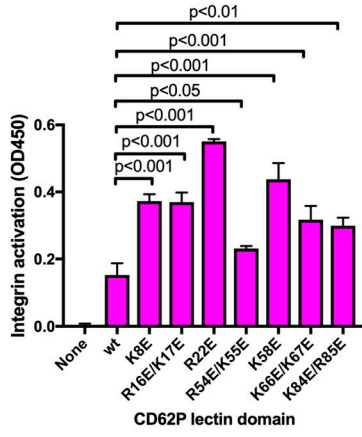

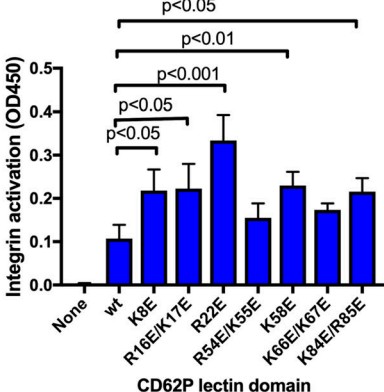

**Figure 4. Activation of integrin αvβ3 by the CD62P lectin domain.**
**(A)** A docking model of the CD62P lectin domain binding to site 2 of αvβ3. Docking simulation was performed as described in the Materials and Methods section. The specificity loop (residues 177–184 of β3) located between site 1 and site 2 is shown. Site 1 is on the opposite side of site 2 (Fig 2B). **(B, C)** Activation of soluble integrins αvβ3 (B) and αIIbβ3 (C) by the CD62P lectin domain in ELISA-type activation assays. Wells of 96-well microtiter plate were coated with ligands (γC399tr for αvβ3 at 50 μg/ml, and γC390-411 for αIIbβ3 at 20 μg/ml) and the remaining protein-binding sites were blocked with BSA. Wells were incubated with soluble integrins (1 μg/ml) and the CD62P lectin domain (0–50 μg/ml) in Tyrode–HEPES buffer with 1 mM Ca²⁺ for 1 h, and bound integrins were quantified using anti-β3 mAbs and HRP-conjugated anti-mouse IgG. Data are shown as means ± SD (n = 3). Statistical analysis was done by ANOVA in Prism7. **(D, E)** Effect of mutations in the predicted site 2-binding interface (Table 2) on

**Table 1.  Amino acid residues in interaction between the CD62P lectin domain (1G1Q.pdb) and αvβ3 (1L5G.pdb) as predicted by the docking simulation.**

| CD62P lectin domain | αv | β3 |
|---|---|---|
| Asn13, Ile14, **Arg16, Lys17**, Tyr18, Gln20, Asn21, Thr24, Asp25, **Lys55**, Asn56, **Lys58**, Thr61, Val63, Gly64, Thr65, **Lys66, Lys67**, Ala68, Asn83, **Lys 84, Arg85**, Asn86, Asn87, Asp89, His108, Leu110, Ala120, Cys122, Gln123, Asp124, Met125, Lys129, Glu132, Cys133, Glu135, Tyr145 | Ala149, Asp150, Phe177,Tyr178, Gln180, Arg211, Thr212, Ala213, Gln214, Ala215, Ile216, Asp218, Arg218, | Tyr122, Ser123, Met124, Lys125, Asp126, Asp127, Trp129, Ser130, Gln132, Asn133, Lys137, Asp179, Met180, Lys181, Thr182, Arg214, Asp251, Ser334, Met335, Asp336, Ser337, Asn339, Val340, Leu341, Gln342 |

Amino acid residues within 0.6 nm between the CD62P lectin domain and αvβ3 were selected using pdb viewer (version 4.1). Amino acid residues in the CD62P lectin domain selected for mutagenesis are shown in bold.

**Table 2.  Amino acid residues in interaction between the CD62P lectin domain (1G1Q.pdb) and αvβ3 (1JV2.pdb) as predicted by the docking simulation.**

| CD62P lectin domain | αv | β3 |
|---|---|---|
| Thr7, **Lys8**, Trp12, Asn13, Ile14, **Arg16, Lys17**, Tyr18, Gln20, Asn21, **Arg22**, Tyr23, Thr24, Asp25, **Arg54, Lys55**, Asn56, Asn57, **Lys58**, Thr59, Val63, Gly64, Thr65, **Lys66, Lys67**, Ala68, Asn86, Asn87, Asp89, Leu110, Tyr118, Thr119, Ala120, Ser121, Cys122, Gln123, Asp124, Met125, Glu135 | Glu15, Asn44, Thr45, Thr46, Pro48, Gly49, Ile50, Val51, Glu52, Gly76, Asn77, Asp79, Asp83, Asp84, Pro85, Phe88, His91, Gln120, Arg122 | Lys159, Pro160, Val161, Met165, Ile167, Ser168, Glu171, Ala172, Glu174, Asn175, Pro176, Cys177, Tyr178, Asp179, Met180, Lys181, Thr183, Cys184, Pro186, Met187, Phe188, **Val275, Gly276, Ser277, Asp278, His280, Tyr281, Ser282, Ser284, Thr285, Thr286,** |

Amino acid residues within 0.6 nm between the CD62P lectin domain and αvβ3 were selected using pdb viewer (version 4.1). Amino acid residues in β3 site 2 peptides are shown in bold. Amino acid residues in the CD62P lectin domain selected for mutagenesis are shown in bold.

Furthermore, the CD62P lectin domain mutant R16E/K17E, which was defective in binding to soluble integrins, was defective in supporting the adhesion of CHO cells. In contrast, the E88D mutant, which is defective in glycan binding, only weakly affected cell adhesion. These findings suggest that the CD62P lectin domain supported cell adhesion in an integrin-dependent manner. We demonstrated that WT CHO cells that express low levels of αvβ3 still adhere to the CD62P lectin domain, in a manner dependent on β1 integrins. The latter findings make it highly likely that cell-surface CD62P interacting with these integrins may play a role in leukocyte recruitment and cancer metastasis. Previous studies showed that several cytokines allosterically activated integrins without inside–out signaling by binding to and inducing the allosteric site (site 2) of integrins to bind cognate ligands. Notably, the docking simulation of interaction between the CD62P lectin domain and the closed headpiece of integrin αvβ3 predicted that the CD62P lectin domain binds to site 2. Consistent with this prediction, the CD62P lectin domain activated soluble integrins αvβ3 and αIIbβ3 in a dose-dependent manner. We also found that the CD62P lectin domain activated cell-surface integrins in cell-based activation assays. The predicted site 2-binding site of the CD62P lectin domain overlaps with the site 1-binding site. We found that the point mutations in the predicted site 2-binding interface showed enhanced integrin activation compared with WT CD62P lectin domain, and we assume that these mutations result in gain-of-function.

### The CD62P lectin domain binds to and activates β1 integrins

The observation that parent CHO cells (αvβ3-low, α5β1+) adhered to the CD62P lectin domain, suggests that integrins other than αvβ3

and αIIbβ3 can bind to the CD62P lectin domain. Supporting this was the observation that soluble α5β1 and α4β1 bound to the CD62P lectin domain. Notably, the CD62P lectin domain binding to α4β1 was not suppressed by known α4β1 antagonists (BIO1211, or a fibronectin fragment H120). Furthermore, we demonstrated that the CD62P lectin domain allosterically activated cell surface α5β1 and α4β1 on CHO cells. The CD62P lectin domain binds to site 2 peptide from β1, indicating that the CD62P lectin domain binds to site 2 of β1 integrins on CHO cells. These findings suggest that CD62P on activated endothelial cells and platelets are expected to interact with integrins to mediate cell–cell adhesion and signal transduction on a variety of cell types, including immune-competent cells and cancer cells.

CD62P is a transmembrane protein and the CD62P lectin domain is highly concentrated on the cell surface. This is consistent with the requirement for relatively high concentrations of the soluble CD62P lectin domain (>10 µg/ml, ~125 nM) to activate integrins in the ELISA or cell-based binding assays. Also, soluble CD62P is known to bind to heparin and can be highly concentrated on the cell surface by binding to cell-surface proteoglycans (Monzavi-Karbassi et al, 2007; Cooney et al, 2011). Thus, the concentrations necessary to activate integrins in the current study is expected to be biologically relevant in vivo in regulating cell adhesion and signaling.

### Implications of P-selectin activation of integrin-mediated blood cell adhesion

Integrins are expressed on a wide variety of cell types, in contrast to PSGL-1, which is limited to leukocytes. Thus, future studies on the role of CD62P lectin domain binding to integrins on a noncanonical

activation of αvβ3 (D) or αIIbβ3 (E). Because site 1-binding (Table 1) and site 2-binding (Table 2) interfaces in the CD62P lectin domain overlap, we used the same mutations that affect binding to site 1 for activation assays. Data are shown as means ± SD (n = 3). Statistical analysis was done by ANOVA in Prism7.

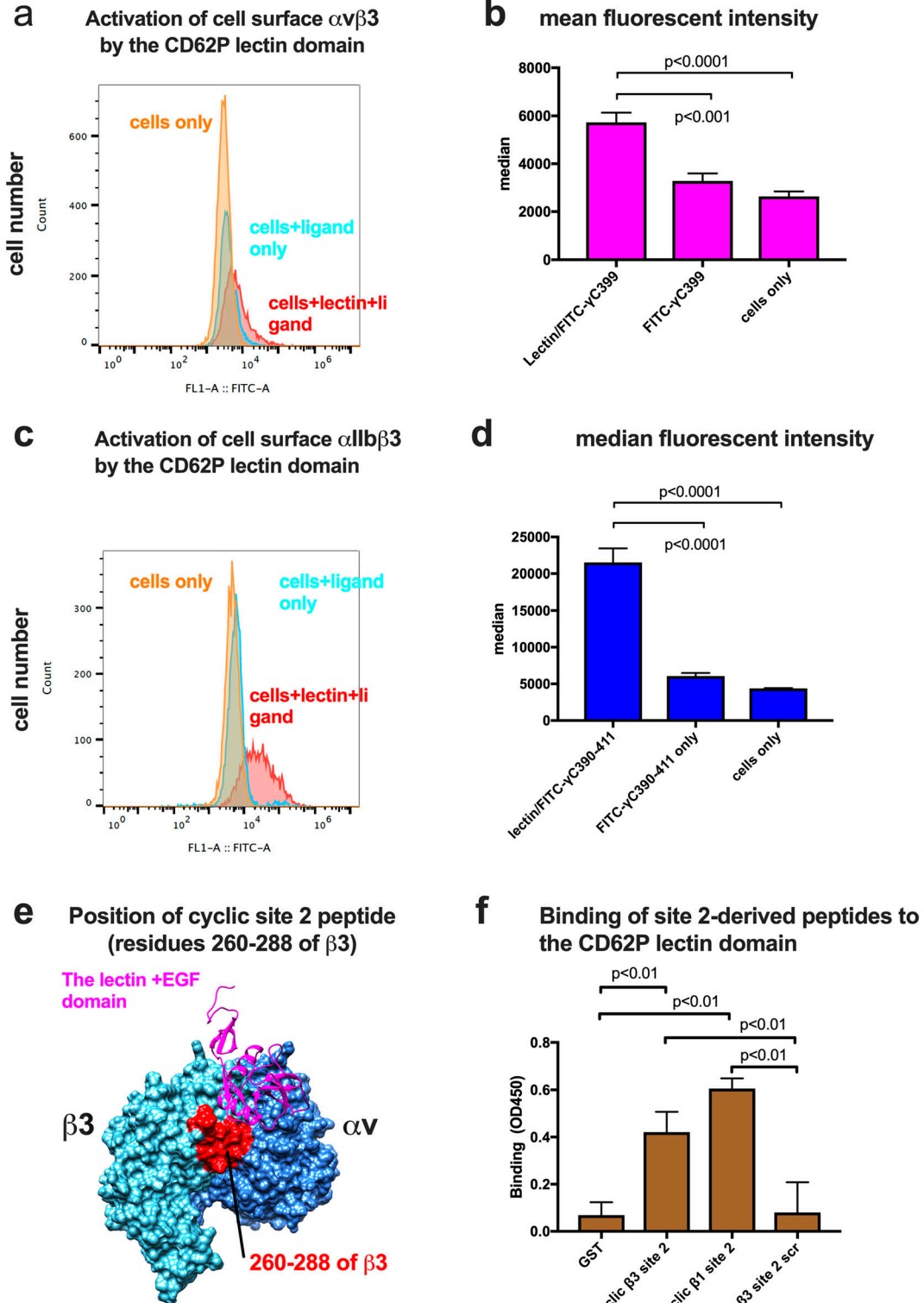

**Figure 5. Activation of cell-surface integrins αvβ3 and αIIbβ3 by the CD62P lectin domain.**
**(A)** Activation of cell-surface αvβ3 on β3-CHO cells by the CD62P lectin domain (flow cytometry). Cells were incubated with FITC-labeled γC399tr in the presence of the CD62P lectin domain (100 μg/ml) and the binding of γC399tr was measured in flow cytometry. **(B)** Median fluorescent intensity (MFI) of the binding of FITC-labeled γC399tr to β3-CHO cells. MFI in flow cytometry was calculated. Data are shown as means ± SD (n = 3). Statistical analysis was done by ANOVA in Prism7. **(C)** Activation of cell-surface αIIbβ3 on αIIbβ3-CHO cells by the CD62P lectin domain (flow cytometry). Cells were incubated with FITC-labeled γC390-411 in the presence of the CD62P lectin domain (100 μg/ml) and the binding of FITC-labeled γC390-411 was measured in flow cytometry. **(D)** MFI of the binding of FITC-labeled γC390-411 to αIIbβ3-CHO cells. MFI in flow cytometry was calculated. Data are shown as means ± SD (n = 3). Statistical analysis was done by ANOVA in Prism7. **(E)** Position of the cyclic site 2 peptide of β3 (residues

cell types may expand our understanding of the true functions of CD62P. It is likely that integrins are maintained in an inactive state on the cell surface in body fluids with high [Ca²⁺]; it may be functionally important that integrins can be activated upon binding to CD62P for supporting cell–cell interaction mediated by CD62P–integrin binding. It has been well established that CD62P on activated endothelial cells binds to PSGL-1 on leukocytes to initiate tethering and rolling during inflammatory recruitment (Springer, 1994). The present study suggests that CD62P on endothelial cells may bind to integrins in addition to binding to glycans (PSGL-1) on the leukocyte membrane. Thus, CD62P binding to site 2 may provide a second mechanism for allosterically activating leukocyte integrins to support shear resistant cell arrest. CD62P is also expressed on activated platelets making it possible that it interacts with integrin αIIbβ3 on the same or opposing platelets. This may lead to direct activation of αIIbβ3 by CD62P binding to site 2 in cis or trans during platelet–platelet interactions.

Previous studies showed that CD62P on activated endothelial cells or activated platelets is involved in tumor metastasis (Weber et al, 2016), but ligands for CD62P on cancer cells have not been fully established. Integrin αvβ3 is known to be overexpressed in many cancers. We hypothesize that CD62P-αvβ3 binding and activation of αvβ3 by CD62P may be involved in metastatic interactions between cancer and endothelial cells.

We conclude that CD62P–integrin interactions present a new therapeutic target in diseases of inflammation and cancer. Because most currently available integrin and selectin antagonists were selected for blocking integrin binding to known extracellular matrix ligands or CD62P binding to glycans, it would be prudent to develop inhibitors of CD62P–integrin site 2 interactions to determine their efficacy in inflammation and cancer.

# Materials and Methods

## Materials

Antibody P8G6 (Santa Cruz Biotechnology), KF38789 (Tocris Bioscience), and PSGL-1-Fc (Sino Biological) were obtained from the described sources.

## The CD62P lectin domain and combined CD62P lectin and the EGF domains

The cDNA fragments encoding the CD62P lectin domain WTYHYSTKAYSWNISRKYCQNRYTDLVAIQNKNEIDYLNKVLPYYSSYY-WIGIRKNNKTWTWVGTKKALTNEAENWADNEPNNKRNNEDCVEIYIKS-PSAPGKWNDEHCLKKKHALC (residues 1–117) were chemically synthesized and subcloned at the BamHI/EcoRI site of pET28a. This protein has N-terminal His tag from PET28a (MGSSHHHHHHSSGLVPRGSHMASMTGGQQMGRGS [molecular weight 3,562]) and the calculated molecular weight is 17,611. The cDNA

fragment encoding the CD62P lectin and the EGF domains (WTYHYSTKAYSWNISRKYCQNRYTDLVAIQNKNEIDYLNKVLPYYSSYY WIGIRKNNKTWTWVGTKKALTNEAENWADNEPNNKRNNEDCVEIYIKS PSAPGKWNDEHCLKKKHALCYTASCQDMSCSKQGECLETIGNYTCSCYPGFYGPE-CEYVRE, residues 1–158) was subcloned into at the BamHI/EcoRI site of PET28a and this protein has N-terminal His tag from PET28a as described above and the calculated molecular weight is 22,210. Protein expression was induced by IPTG in *E. coli* BL21 and purified in Ni-NTA-affinity chromatography under denaturing conditions and refolded as described Fujita et al (2012). SDS–PAGE of the purified protein is shown in Fig S1A.

## GST fusion protein of cyclic site 2 peptides

We synthesized cDNA encoding (a) residues 260–288 of β3 CRLA-GIVQPNDGQSHVGSDNHYSASTTMC (C273 is changed to S); (b) residues 268–295 of β1 (C281 is changed to S) CKLGGIVLPNDGQSHLENNMYTM-SHYYC; and (c) a scrambled site 2 peptide of β3 (VHDSHYSGQ-GAMSDNTNSPQT). They were subcloned into the BamHI/EcoRI site of pGEX-2T6His (Fujita et al, 2014). Protein expression was induced by IPTG in *E. coli* BL21 and purified in glutathione affinity chromatography.

## The disintegrin domain of ADAM15

The disintegrin domain of ADAM15 was synthesized as GST fusion proteins as described (Zhang et al, 1998). Fibrinogen γ-chain C-terminal residues 390–411 cDNA encoding (6 His tag)[HHHHHH] NRLTIGEGQQHHLGGAKQAGDV] was conjugated with the C-terminus of GST (designated γC390-411) in pGEXT2 vector (BamHI/EcoRI site). The protein was synthesized in *E. coli* BL21 and purified using glutathione affinity chromatography. CHO cells that express recombinant human αIIbβ3 were described Kamata et al (1996). The truncated fibrinogen γ-chain C-terminal domain (γC399tr) was generated as previously described Yokoyama et al (1999).

## Binding of soluble integrins to the CD62P lectin domain

ELISA-type binding assays were performed as described previously Fujita et al (2012). Briefly, wells of 96-well Immulon 2 microtiter plates (Dynatech Laboratories) were coated with 100 μl PBS containing the CD62P lectin domain for 2 h at 37°C. Remaining protein-binding sites were blocked by incubating with PBS/0.1% BSA for 30 min at room temperature. After washing with PBS, soluble recombinant αIIbβ3 or αvβ3 (AgroBio, 1 μg/ml) was added to the wells and incubated in HEPES–Tyrodes buffer (10 mM HEPES, 150 mM NaCl, 12 mM NaHCO₃, 0.4 mM NaH₂PO₄, 2.5 mM KCl, 0.1% glucose, 0.1% BSA) with 1 mM MnCl₂ for 1 h at room temperature. After unbound αIIbβ3 was removed by rinsing the wells with binding buffer, bound αIIbβ3 was measured using anti-integrin β3 mAb (AV-10, a kind gift from Brunie Felding, The Scripps Research Institute) followed by HRP-conjugated goat anti-mouse IgG and

260–288 of β3). **(F)** Binding of site 2 peptides from β1 and β3 to the CD62P lectin domain. Wells of the 96-well microtiter plate were coated with the CD62P lectin domain (20 μg/ml) and remaining protein-binding sites were blocked with BSA. Wells were incubated with site 2 peptides fused to GST (100 μg/ml) and bound GST was quantified using HRP-conjugated anti-GST.

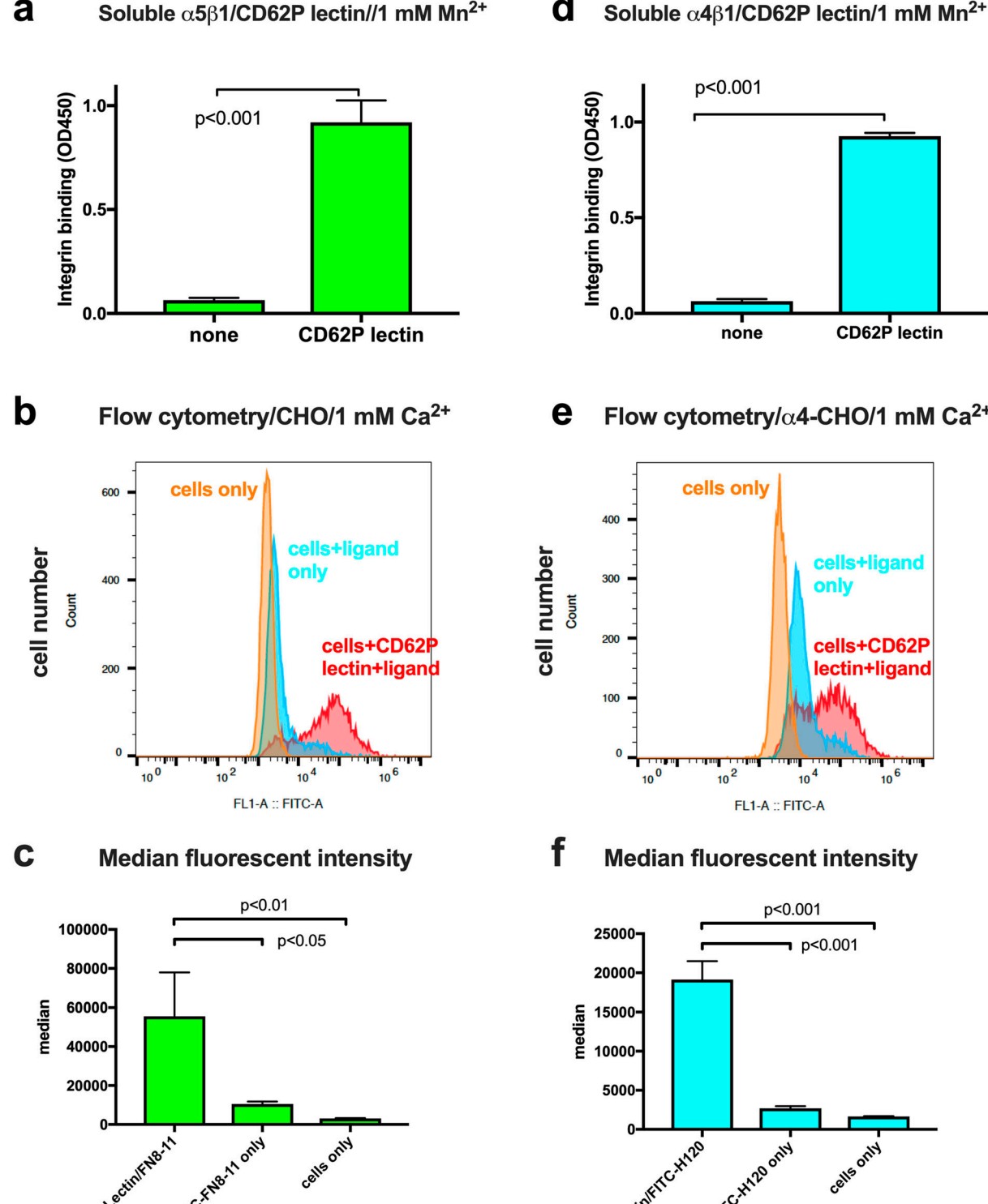

**Figure 6. The CD62P lectin domain binds to and activate integrins $\alpha5\beta1$ and $\alpha4\beta1$.**
**(A)** The binding of biotinylated $\alpha5\beta1$ to immobilized CD62P lectin domain was determined in 1 mM $Mn^{2+}$ as described in Fig 1 except that bound integrin was quantified using streptavidin conjugated with HRP. **(B)** Activation of cell-surface $\alpha5\beta1$ on CHO cells by the CD62P lectin domain (flow cytometry). Cells were incubated with FITC-labeled fibronectin cell-binding domain (FN8-11) in the presence of the CD62P lectin domain (50 $\mu$g/ml) and the binding of FN8-11 was measured in flow cytometry. **(C)** Median fluorescent intensity (MFI). MFI in flow cytometry was calculated. **(D)** The binding of the CD62P lectin domain to integrins $\alpha4\beta1$. The binding of biotinylated $\alpha4\beta1$ to immobilized CD62P lectin domain was determined in 1 mM $Mn^{2+}$ as described in Fig 1 except that bound integrin was quantified using streptavidin conjugated with HRP. **(E)** Activation of cell-surface $\alpha4\beta1$ on CHO cells by the CD62P lectin domain (flow cytometry). Cells were incubated with FITC-labeled fibronectin $\alpha4\beta1$-binding fragment (H120) in the presence of the CD62P lectin domain (50 $\mu$g/ml) and the binding of H120 was measured in flow cytometry. **(F)** MFI in flow cytometry was calculated.

peroxidase substrates. The binding of soluble $\alpha4\beta1$ or $\alpha5\beta1$ was performed as described above except that biotinylated $\alpha4\beta1$ or $\alpha5\beta1$ (AgroBio) were used. The binding of biotinylated integrins was measured using HRP-conjugated streptavidin and 3'3'5'5'-tetramethylbenzidine (TMB) as a substrate.

## Activation of soluble integrins by the lectin domain

ELISA-type binding assays were performed as described previously Fujita et al (2018). Briefly, wells of 96-well Immulon 2 microtiter plates were coated with 100 $\mu$l PBS containing $\gamma$C399tr (for $\alpha v\beta3$) or $\gamma$C390-411 (for $\alpha IIb\beta3$) for 2 h at 37°C. The remaining protein-binding sites were blocked by incubating with PBS/0.1% BSA for 30 min at room temperature. After washing with PBS, soluble recombinant $\alpha IIb\beta3$ or $\alpha v\beta3$ (AcroBio, 1 $\mu$g/ml) was preincubated with the CD62P lectin domain for 10 min at room temperature and was added to the wells and incubated in HEPES–Tyrodes buffer with 1 mM $CaCl_2$ for 1 h at room temperature. After unbound integrins were removed by rinsing the wells with the binding buffer, bound integrins were measured using anti-integrin $\beta3$ mAb (AV-10) followed by HRP-conjugated goat anti-mouse IgG and peroxidase substrates. Activation of $\alpha4\beta1$ or $\alpha5\beta1$ was measured as described above except that the FN H120 fragment (specific to $\alpha4\beta1$) fused to GST or FN 8–11 fragment (specific to $\alpha5\beta1$) fused to GST were used as ligands and the binding of biotinylated integrins was measured using HRP-conjugated streptavidin.

## Activation of cell surface integrins by the CD62P lectin domain

For activation of $\alpha IIb\beta3$ on $\alpha IIb\beta3$-CHO, cells were cultured in DMEM/10% FCS. The cells were resuspended with HEPES–Tyrodes buffer/0.02% BSA (heat-treated at 80°C for 20 min to remove contaminating cell adhesion molecules). The cells were then incubated with CD62P lectin domain for 30 min on ice and then incubated with FITC-labeled $\gamma$C390-411 (50 $\mu$g/ml) for 30 min at room temperature. The $\alpha IIb\beta3$-CHO cells were washed with PBS/0.02% BSA and analyzed in BD Accuri flow cytometer (Becton Dickinson). For activation of $\alpha v\beta3$ on $\beta3$-CHO cells, we used the same protocol except that $\beta3$-CHO cells and FITC-labeled $\gamma$C399tr were used. The data were analyzed using FlowJo 7.6.5.

## Adhesion of CHO cells (PSGL-1 negative) to the CD62P lectin domain

Wells of 96-well microtiter plate were coated with the CD62P lectin domain (WT and mutants, coating concentration at 50 $\mu$g/ml) and the remaining protein-binding sites were blocked with BSA. Wells were incubated with CHO cells ($\alpha5\beta1$+) in Tyrode–HEPES/1 mM $Mg^{2+}$ or DMEM for 1 h at room temperature and the adherent CHO was quantified using endogenous phosphatase activity using p-nitrophenyl phosphate as a substrate as previously described Kamata et al (2002).

## Docking simulation

Docking simulation of interaction between CD62P (PDB code 1G1Q), and integrin $\alpha v\beta3$ was performed using AutoDock3, as described

Ieguchi et al (2010). In the current study, we used the headpiece (residues 1–438 of $\alpha v$ and residues 55–432 of $\beta3$) of $\alpha v\beta3$ (PDB code 1L5G, open headed). Cations were not present in $\alpha v\beta3$ during docking simulation (Mori et al, 2008; Saegusa et al, 2008). The classical ligand-binding site (site 1) or the allosteric site (site 2) of $\alpha v\beta3$ was selected as a target for the lectin domain. To perform docking simulation of the interaction between site 2 of closed headed $\alpha v\beta3$, we used 1JV2.pdb. Site-directed mutagenesis was carried out as previously described Takada et al (2019).

## Statistical analysis

Treatment differences were tested using ANOVA and a Tukey multiple comparison test to control the global type I error using Prism 7 (GraphPad Software).

# Supplementary Information

# Acknowledgements

This project was supported by pilot funding from the Comprehensive Cancer Center at UC Davis School of Medicine. This work is partly supported by the UC Davis Comprehensive Cancer Center Support Grant (CCSG) awarded by the National Cancer Institute (NCI P30CA093373) and R01 AI407294 to SI Simon.

## Author Contributions

YK Takada: data curation.
SI Simon: resources and writing—review and editing.
Y Takada: conceptualization, formal analysis, funding acquisition, and project administration.

## Conflict of Interest Statement

The authors declare that they have no conflict of interest.

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
