## [Reviewer comments · Life Science Alliance]

Life Science Alliance

The C-type lectin domain of CD62P (P-selectin) functions as an integrin ligand.

Yoko Takada, Scott Simon, and Yoshikazu Takada

DOI: <https://doi.org/10.26508/lsa.202201747>

Corresponding author(s): Yoshikazu Takada, UC Davis

Review Timeline:

Submission Date:	2022-09-29
Editorial Decision:	2022-11-09
Revision Received:	2023-02-15
Editorial Decision:	2023-03-14
Revision Received:	2023-04-06
Editorial Decision:	2023-04-11
Revision Received:	2023-04-14
Accepted:	2023-04-17

Scientific Editor: Novella Guidi

Transaction Report:

November 9, 2022

Re: Life Science Alliance manuscript #LSA-2022-01747-T

Dr. yoshikazu takada
UC Davis
Dermatology, Biochemistry and Molecular Medicine
4645 2nd ave
4645 Second Avenue
sacramento, CA 95817

Dear Dr. takada,

Thank you for submitting your manuscript entitled "The C-type lectin domain of CD62P (P-selectin) is an integrin ligand." to Life Science Alliance. The manuscript was assessed by expert reviewers, whose comments are appended to this letter. We invite you to submit a revised manuscript addressing the Reviewer comments.

Thank you for this interesting contribution to Life Science Alliance. We are looking forward to receiving your revised manuscript.

Sincerely,

B. MANUSCRIPT ORGANIZATION AND FORMATTING:

Reviewer #1 (Comments to the Authors (Required)):

This manuscript by Y.K. Takada and Y. Takada reports for the first time that the C-type lectin domain of P-selectin (CD62P) can act as a ligand of different $\beta 3$ ($\alpha 11\beta 3$ and $\alpha V\beta 3$) and $\beta 1$ ($\alpha 4\beta 1$ and $\alpha 5\beta 1$) integrins, thereby supporting integrin-mediated cell adhesion. Through mutational analyses, the authors present data indicating that the binding sites on the CD62P lectin domain for glycan (its "classical ligand") and integrin are distinct. Furthermore, authors claim that CD62P lectin domain can bind to the "classical" ligand-binding site ("site 1") of integrin molecules, but it also binds to the allosteric ("site 2") site on the integrin head part, thereby inducing integrin activation.

The design and execution of the experimental approaches are in general terms correct and the results presented in the manuscript are interesting and represent a contribution towards a broader understanding of the involvement of P-selectin in cell-cell interactions beyond its well-known role in binding sialyl-Lewis X decorated glycoproteins. However, the following major points should be properly addressed in order to enhance the interest of the paper for a broader audience, as well as its soundness:

1. An important claim made by authors is that the lectin domain of P-selectin, by binding to the allosteric integrin site (site 2), induces integrin activation. This important conclusion is only weakly supported by the data shown in Figure 5. In fact, many other interpretations could explain the results in this Figure, including that the lectin domain could be acting as a bridge by binding simultaneously to the fibrinogen fragments and integrins. In order to demonstrate that the lectin domain can bind to integrin site 2, authors should mutate relevant residues in integrin site 2 and show that this abrogates integrin activation (as evidenced by increased binding to immobilized fibrinogen fragments) induced by the lectin domain of P-selectin.
2. Some assertions in the manuscript are clearly overstated; for instance the sentence (at the beginning of Discussion): "Notably, the present study defines the role of integrins in CD62P-mediated cell-cell interaction in the pathogenesis of diseases" is clearly overstated as the manuscript provides absolutely no data regarding disease pathogenesis.
3. In order to better illustrate the role of divalent cations (Ca^{2+} , Mg^{2+} and Mn^{2+}) for integrin ligand binding, the Introduction section should be expanded and include a more detailed description of integrin "site 1" including the location of MIDAS, ADMIDAS and LIMBS motifs. The precise location of these important motifs should also be shown in Figure 2a and 2b. Similarly, a better description of the synergistic integrin "site 2" should be included in the Introduction and its precise location shown in Figure 5a.
4. The Discussion should be expanded to better accommodate all the results reported in the manuscript. Figure 6 and the corresponding legend could be removed, as it does not provide any additional information compared to what is described in the Discussion.

Minor points:

1. What is CD450 in the y-axis in Figs 1c, 2c, 2d and 5b ? Should it not be OD450 (optical density at 450 nm)?
2. The residues K16 and R17 depicted in blue in Figure 2a and 2b are incorrect. The correct ones should be R16 and K17. Similarly, the residue R84 shown in Figure 2b is also incorrect (should be K84).
3. There are many typos and grammatical mistakes throughout the manuscript. For instance many verbs in third-person singular lack the final "s". In addition some sentences simply make no sense at all; these are just a few examples:
"To test this possibility, we used ELISA-type activation assays, in which soluble integrins $\alpha V\beta 3$ and $\alpha 11\beta 3$ were incubated with immobilized fibrinogen fragments, $\gamma C399\text{tr}$ and $\gamma C390-411$ specific to $\alpha V\beta 3$ and $\alpha 11\beta 3$, respectively, and incubated with soluble integrins in the presence of the lectin domain in 1 mM Ca^{2+} (to keep integrins inactive)."

"The model predicts that PSGL-1 peptide (605YEYLDYDFLPETEP618) in the PSGL-1-CD62P lectin domain complex (1g1s.pdb) (13) binds to CD62P to integrin $\alpha V\beta 3$ without steric hindrance (Fig. 2a)."

I strongly recommend that authors seek a professional English editing service in order to enhance the grammatical coherence of the manuscript and correct all the spelling mistakes and typos.

Reviewer #2 (Comments to the Authors (Required)):

By an *in silico* approach the authors predict that CD62P, a selectin exposed on the surface of activated platelets and endothelial cells, is a ligand for integrins able to mediate heterotypic cell-to-cell adhesion. They confirmed this prediction by showing that the lectin domain is required to bind integrin $\alpha\text{IIb}\beta\text{3}$ and $\alpha\text{v}\beta\text{3}$ and that the known integrin ligand ADAM15 competes for CD62P. Interestingly they also suggest that CD62P increases integrin activation. The results shown are interesting but more controls are required as well as insights to demonstrate the biological relevance.

CRITICISMS

Figure 1. The authors have to evaluate by Scatchard analysis the affinity of the interaction of the lectin domain with $\alpha\text{v}\beta\text{3}$. Do other $\alpha\text{v}\beta\text{3}$ natural ligands (vitronectin, fibronectin) and RGD peptidomimetics compete with CD62P lectin domain binding?

As suggested by Figure 2e, it is likely that the glycan and integrin binding sites may interfere each other because very close. Is there any cross competition between the two ligands?

Figure 3. I suggest to show that the CHO cells exploited really do not express PSGL-1 by FACS analysis. Furthermore experiments on cells (e.g. endothelial cells) that express both PSGL-1 and $\alpha\text{v}\beta\text{3}$ are more relevant. It is important to analyze the binding capacity of endothelial cells genetically manipulated to block/reduce the expression of $\alpha\text{v}\beta\text{3}$ and/or PSGL-1 to lectin domain of CD62P and its mutants

The experiments shown in figure 5 suggest that CD62P lectin domain allows integrin activation. The availability of Ab that distinguish the active and inactive form of $\alpha\text{v}\beta\text{3}$ might better support the data shown. Because integrins mediate outside-in signaling leading to cell spreading I suggest to exploit an easy experiment of endothelial cell adhesion to vitronectin the natural ligand of $\alpha\text{v}\beta\text{3}$ in presence or absence of the lectin domain of CD62P

MINOR CRITICISM

This sentence could be rephrased because not clear enough: " To test this possibility, we used ELISA-type activation assays, in which soluble integrins $\alpha\text{v}\beta\text{3}$ and $\alpha\text{IIb}\beta\text{3}$ were incubated with immobilized fibrinogen fragments, γC399tr and $\gamma\text{C390-411}$ specific to $\alpha\text{v}\beta\text{3}$ and $\alpha\text{IIb}\beta\text{3}$, respectively, and incubated with soluble integrins in the presence of the lectin domain in 1 mM Ca^{2+} (to keep integrins inactive)"

Reviewer #3 (Comments to the Authors (Required)):

In this manuscript the authors propose CD62P (P-selectin) as novel ligand of $\alpha\text{v}\beta\text{3}$, $\alpha\text{IIb}\beta\text{3}$, $\alpha\text{4}\beta\text{1}$ and $\alpha\text{5}\beta\text{1}$ integrins. They identified mutations that inhibit binding of the CD62P C-type lectin domain to integrins and are located outside the glycan binding site. Interestingly, the CD62P lectin domain promoted cell adhesion and activated $\alpha\text{v}\beta\text{3}$ integrins *in vitro*.

The study is well written and follows previous studies by the Takada laboratory which report the identification of new integrin ligands. It mainly uses biochemical assays - *in vitro* binding and activation studies - supplemented by adhesion assays using CHO cells.

The study, although potentially interesting, remains preliminary. The authors require high levels of CD62P C-type lectin domain to detect interaction and activation with integrins *in vitro*. Can the CD62P C-type lectin domain activate integrins on cells? Some functional assays in cells to demonstrate the functional relevance of the *in vitro* data (in addition to the adhesion assays) would have increased the significance of their findings. Nevertheless, I can recommend publication of this study given the successful completion of a number of major and minor issues that would need revision:

1. The method section does not provide sufficient information about the experimental procedures to allow the reproduction of the protocols or to fully evaluate the data. Examples are ADAM15 competition experiments: How was ADAM15 purified, what amount (in nmol) was used for competition experiments? How was the site-directed mutagenesis carried out? How much of the CD62P lectin domain was used (in nmol) to coat the ELISA plates? From which source did the authors obtain soluble $\alpha\text{v}\beta\text{3}$ integrin? What are the sources of anti-integrin β3 mAb (AV-10) - how much was used of this antibody (and HRP-conjugated goat anti-mouse IgG) for the binding assay? What peroxidase substrate has been used? How was the binding assay of CD62P C-type lectin domain to $\alpha\text{4}\beta\text{1}$ and $\alpha\text{5}\beta\text{1}$ carried out? What was the source of these integrins? Adhesion assays using CHO cells - how long were cells incubated on the lectin ligands? What is the source of the fibrinogen fragments,... and so on.
2. Purified C-type lectin domain or the combined C-type lectin and EGF-like domain of CD62P were expressed in bacteria and purified. How was the quality control of the purified proteins controlled (by mass spec)? The authors should include an SDS-PAGE to show the purity and size of the final products.
3. Is it possible to increase the concentration of CD62P C-type lectin domain in their integrin binding assays (Figure 1b) to reach saturation binding and to be able to determine the K_d of binding?
4. How strong is the adhesion to the lectin domain when compared to another $\alpha\text{5}\beta\text{1}$ integrin ligand such as fibronectin? How

long were CHO cells incubated on lectin domain ligands to assay their adhesion? The authors should show two time points (for fast <30 min and slow adhesion <45 min) and include another $\alpha 5\beta 1$ integrin ligand such as fibronectin as control.

5. The activation assay (Figure 5) are interesting and important but lack controls. The authors should prepare a concentration range of CD62P C-type lectin domain (to see if and when the signal reaches saturation) and include their integrin-binding deficient CD62P C-type lectin domain mutants.

Activation was only analyzed in ELISA-type activation assays but not on cells. Is the CD62P C-type lectin domain also able to activate cell surface $\alpha v\beta 3$ and $\alpha IIb\beta 3$ integrins on living cells? What about the activation of $\alpha 4\beta 1$ and $\alpha 5\beta 1$ integrins by the CD62P C-type lectin domain?

6. The authors state that "cRGFfV or 7E3 (anti- $\beta 3$) did not affect the binding of the lectin domain to soluble $\alpha v\beta 3$." However, this data is not shown. There is another instance of "data not shown" in case of inhibitors of CD62P-PSGL-1 interaction. These data should be provided, eventually as supplemental information.

Minor points:

1. Figure 1D: It is written "P-selectin" in the figure but probably only the CD62P lectin domains have been used as in the other panels. The description should be changed.
2. Unclear sentence: "The model predicts that PSGL-1 peptide in the PSGL-1-CD62P lectin domain complex bind to CD62P to integrin $\alpha v\beta 3$ without steric hindrance (Fig. 2a)".
3. Figure 2: How do the authors explain increased binding of R54K/K55E to $\alpha v\beta 3$ and $\alpha IIb\beta 3$ integrin?
4. Why is Figure 4 described before Figure 3 in the result section?

Thank you for the constructive comments and encouragement. We revised the manuscript extensively and added new results to respond to the criticism. Since we were not able to work for over 2 weeks in January because of COVID-19 infection, it took a little more than 3 months. We hope that the revised manuscript is acceptable.

Reviewer #1 (Comments to the Authors (Required)):

This manuscript by Y.K. Takada and Y. Takada reports for the first time that the C-type lectin domain of P-selectin (CD62P) can act as a ligand of different $\beta 3$ ($\alpha 11\beta 3$ and $\alpha V\beta 3$) and $\beta 1$ ($\alpha 4\beta 1$ and $\alpha 5\beta 1$) integrins, thereby supporting integrin-mediated cell adhesion. Through mutational analyses, the authors present data indicating that the binding sites on the CD62P lectin domain for glycan (its "classical ligand") and integrin are distinct. Furthermore, authors claim that CD62P lectin domain can bind to the "classical" ligand-binding site ("site 1") of integrin molecules, but it also binds to the allosteric ("site 2") site on the integrin head part, thereby inducing integrin activation.

The design and execution of the experimental approaches are in general terms correct and the results presented in the manuscript are interesting and represent a contribution towards a broader understanding of the involvement of P-selectin in cell-cell interactions beyond its well-known role in binding sialyl-Lewis X decorated glycoproteins. However, the following major points should be properly addressed in order to enhance the interest of the paper for a broader audience, as well as its soundness:

1. An important claim made by authors is that the lectin domain of P-selectin, by binding to the allosteric integrin site (site 2), induces integrin activation. This important conclusion is only weakly supported by the data shown in Figure 5. In fact, many other interpretations could explain the results in this Figure, including that the lectin domain could be acting as a bridge by binding simultaneously to the fibrinogen fragments and integrins. In order to demonstrate that the lectin domain can bind to integrin site 2, authors should mutate relevant residues in integrin site 2 and show that this abrogates integrin activation (as evidenced by increased binding to immobilized fibrinogen fragments) induced by the lectin domain of P-selectin.

Response:

#1) **Site 2 peptides.** We previously published that peptides from site 2 (residues 267-287) bound to several site 2 ligands (e.g., CX3CL1, CXCL12, CCL5, sPLA2-IIA, and CD40L) and this proves that they bind to site 2. We added the results that cyclic site 2 peptides of b1 and b3 (more stable than linear peptides) bound to the lectin domain (new Fig. 5f), indicating that the lectin domain binds to site 2.

#3) The requested experiments that introduce mutations to the site 2 will be difficult because site 2 contains so many amino acid residues and is probably beyond the scope of the current study.

2. Some assertions in the manuscript are clearly overstated; for instance the sentence (at the beginning of Discussion): "Notably, the present study defines the role of integrins in CD62P-mediated cell-cell interaction in the pathogenesis of diseases" is clearly overstated as the manuscript provides absolutely no data regarding disease pathogenesis.

Response: We agree that this statement is not supported by the data presented and have revised the statement:

3. In order to better illustrate the role of divalent cations (Ca^{2+} , Mg^{2+} and Mn^{2+}) for integrin ligand binding, the Introduction section should be expanded and include a more detailed description of integrin "site 1" including the location of MIDAS, ADMIDAS and LIMBS motifs. The precise location of these important motifs should also be shown in Figure 2a and 2b. Similarly, a better description of the synergistic integrin "site 2" should be included in the Introduction and its precise location shown in Figure 5a.

Response: We appreciate this suggestion to elucidate the description of the various motifs associated with ligand binding and now include a description of site 1, and MIDAS, ADMIDAS, LIMBS, and the specificity loop in the introduction. We included the specificity loop in Figs to show the positions of site 1 and site 2 easier. We showed site 1 and site 2 are on the opposite side of the specificity loop.

4. The Discussion should be expanded to better accommodate all the results reported in the manuscript. Figure 6 and the corresponding legend could be removed, as it does not provide any additional information compared to what is described in the Discussion.

Response: Thanks for the suggestion, in agreement we have deleted Fig. 6 and its legend. We expanded the discussion to include all the reported results.

Minor points:

1. What is CD450 in the y-axis in Figs 1c, 2c, 2d and 5b ? Should it not be OD450 (optical density at 450 nm)? Corrected.

2. The residues K16 and R17 depicted in blue in Figure 2a and 2b are incorrect. The correct ones should be R16 and K17. Similarly, the residue R84 shown in Figure 2b is also incorrect (should be K84). Corrected.

3. There are many typos and grammatical mistakes throughout the manuscript. For instance many verbs in third-person singular lack the final "s". In addition some sentences simply make no sense at all; these are just a few examples:

"To test this possibility, we used ELISA-type activation assays, in which soluble integrins $\alpha\text{v}\beta\text{3}$ and $\alpha\text{IIb}\beta\text{3}$ were incubated with immobilized fibrinogen fragments, γC399tr and $\gamma\text{C390-411}$ specific to $\alpha\text{v}\beta\text{3}$ and $\alpha\text{IIb}\beta\text{3}$, respectively, and incubated with soluble integrins in the presence of the lectin domain in 1 mM Ca^{2+} (to keep integrins inactive)."

"The model predicts that PSGL-1 peptide (605YEYLDYDFLPETEP618) in the PSGL-1-CD62P lectin domain complex (1g1s.pdb) (13) binds to CD62P to integrin $\alpha\text{v}\beta\text{3}$ without steric hindrance (Fig. 2a)."

Corrected.

I strongly recommend that authors seek a professional English editing service in order to enhance the grammatical coherence of the manuscript and correct all the spelling mistakes and typos.

Response: My colleague, Dr Simon, an expert in selectins and integrins, edited the manuscript.

Reviewer #2 (Comments to the Authors (Required)):

By an in silico approach the authors predict that CD62P, a selectin exposed on the surface of activated platelets and endothelial cells, is a ligand for integrins able to mediate heterotypic cell-to-cell adhesion. They confirmed this prediction by showing that the lectin domain is required to bind integrin $\alpha\text{IIb}\beta\text{3}$ and $\alpha\text{v}\beta\text{3}$ and that the known integrin ligand ADAM15 competes for CD62P. Interestingly they also suggest that CD62P increases integrin activation. The results shown are interesting but more controls are required as well as insights to demonstrate the biological relevance.

CRITICISMS

Figure 1. The authors have to evaluate by Scatchard analysis the affinity of the interaction of the lectin domain with $\alpha\text{v}\beta\text{3}$. Do other $\alpha\text{v}\beta\text{3}$ natural ligands (vitronectin, fibronectin) and RGD peptidomimetics compete with CD62P lectin domain binding?

Response: We showed that the ADAM-15 disintegrin domain, a specific ligand for $\alpha\text{v}\beta\text{3}$ and $\alpha\text{IIb}\beta\text{3}$, inhibited CD62P lectin domain binding. Other known inhibitors did not inhibit the binding (supplemental Fig S1). This may be because currently available inhibitors are optimized using ECM ligands, not CD62P lectin domain. Also, we propose that the lectin domain possibly binds to the classical binding site (site 1) and the allosteric site (site 2) (two binding site model). If this is the case, binding kinetics would be difficult to analyze, unless we have two different ligands that bind to site 1 and site 2 separately. We will address this issue in future experiments.

As suggested by Figure 2e, it is likely that the glycan and integrin binding sites may interfere each other because very close. Is there any cross competition between the two ligands?

Response: As shown in supplemental Fig. S1 several known antagonists for glycan binding did not inhibit CD62P lectin domain binding to integrins. Thus, it is unclear if the lectin binding site is associated with integrin binding.

Figure 3. I suggest to show that the CHO cells exploited really do not express PSGL-1 by FACS analysis. Furthermore experiments on cells (e.g. endothelial cells) that express both PSGL-1 and $\alpha\text{v}\beta\text{3}$ are more relevant. It is important to analyze the binding capacity of endothelial cells genetically manipulated to block/reduce the expression of $\alpha\text{v}\beta\text{3}$ and/or PSGL-1 to lectin domain of CD62P and its mutants

Response: We could not perform the suggested experiments. since commercial anti-PSGL-1 antibodies are all specific to human or mouse PSGL-1. Further, there is not a commercial antibody recognizes hamster PSGL-1 and it is known that CHO cells lack two enzymes (i.e. core 2 beta1-6-N-acetylglucosaminyltransferase (C2GnT) and an alpha1-3 fucosyltransferase) that mediate branching and fucosylation of the O-linked glycan modifications. Therefore, even if CHO cells synthesize PSGL-1 core protein by transfection of human PSGL-1 cDNA, PSGL-1 core protein would not bind to CD62P (Li et al 1996, JBC).

The experiments shown in figure 5 suggest that CD62P lectin domain allows integrin activation. The availability of Ab that distinguish the active and inactive form of $\alpha\text{v}\beta\text{3}$ might better support the data shown.

Response: We previously published that site 2-mediated allosteric integrin activation did not accompany global conformational changes, and we did not detect the exposure of epitopes for conformation-specific antibodies (Fujita PLOSone 2014, Fujita JBC 2015).

Because integrins mediate outside-in signaling leading to cell spreading I suggest to exploit an easy experiment of endothelial cell adhesion to vitronectin the natural ligand of $\alpha v\beta 3$ in presence or absence of the lectin domain of CD62.

Response: Endothelial cells express many proteins on the surface, including CX3CL1 and endogenous CD62P. We have previously reported that CX3CL1 binds to integrins $\alpha IIb\beta 3$ and $\alpha v\beta 3$ and allosterically activate integrins as well (Fujita JI 2012). Since endothelial cells are complex, we instead employed recombinant integrins $\alpha v\beta 3$, $\alpha IIb\beta 3$, $\alpha 5\beta 1$, and $\alpha 4\beta 1$ on CHO cells to demonstrate activation by CD62 lectin domain. We showed that the lectin domain activated cell surface integrins using flow cytometry (new Figs. 5 and 6)

MINOR CRITICISM

This sentence could be rephrased because not clear enough: " To test this possibility, we used ELISA-type activation assays, in which soluble integrins $\alpha v\beta 3$ and $\alpha IIb\beta 3$ were incubated with immobilized fibrinogen fragments, $\gamma C399tr$ and $\gamma C390-411$ specific to $\alpha v\beta 3$ and $\alpha IIb\beta 3$, respectively, and incubated with soluble integrins in the presence of the lectin domain in 1 mM Ca^{2+} (to keep integrins inactive)"

Response. We revised the sentence. "To test this possibility, we used ELISA-type activation assays. Wells of 96-well microtiter wells were coated with fibrinogen fragments, $\gamma C399tr$ and $\gamma C390-411$ specific to $\alpha v\beta 3$ and $\alpha IIb\beta 3$, respectively. We incubated the wells with soluble integrins $\alpha v\beta 3$ and $\alpha IIb\beta 3$ in the presence of the lectin domain in 1 mM Ca^{2+} (to keep integrins inactive)."

Reviewer #3 (Comments to the Authors (Required)):

In this manuscript the authors propose CD62P (P-selectin) as novel ligand of $\alpha v\beta 3$, $\alpha IIb\beta 3$, $\alpha 4\beta 1$ and $\alpha 5\beta 1$ integrins. They identified mutations that inhibit binding of the CD62P C-type lectin domain to integrins and are located outside the glycan binding site. Interestingly, the CD62P lectin domain promoted cell adhesion and activated $\alpha v\beta 3$ integrins in vitro.

The study is well written and follows previous studies by the Takada laboratory which report the identification of new integrin ligands. It mainly uses biochemical assays - in vitro binding and activation studies - supplemented by adhesion assays using CHO cells.

The study, although potentially interesting, remains preliminary. The authors require high levels of CD62P C-type lectin domain to detect interaction and activation with integrins in vitro. Can the CD62P C-type lectin domain activate integrins on cells? Some functional assays in cells to demonstrate the functional relevance of the in vitro data (in addition to the adhesion assays) would have increased the significance of their findings. Nevertheless, I can recommend publication of this study given the successful completion of a number of major and minor issues that would need revision:

Response: We added the data that cell surface integrins are activated by the lectin domain (New Fig. 5 and 6).

1. The method section does not provide sufficient information about the experimental procedures to allow the reproduction of the protocols or to fully evaluated the data. Examples are ADAM15 competition experiments: How was ADAM15 purified, what amount (in nmol) was used for competition experiments? How was the site-directed mutagenesis carried out? How much of the CD62P lectin domain was used (in nmol) to coat the ELISA plates? From which source did the authors obtain soluble $\alpha v\beta 3$ integrin? What are the sources of anti-integrin $\alpha IIb\beta 3$ mAB (AV-10) - how much was used of this antibody (and HRP-conjugated goat anti-mouse IgG) for the binding assay? What peroxidase substrate has been used? How was the binding assay of CD62P C-type lectin domain to $\alpha 4\beta 1$ and $\alpha 5\beta 1$ carried out? What was the source of these integrins? Adhesion assays using CHO cells - how long were cells incubated on the lectin ligands? What is the source of the fibrinogen fragments,... and so on.

Response: We added the missing information. We included the molecular weight of the proteins to make it easier to convert the data to nmol. We included the amount of antibodies used in the legend.

2. Purified C-type lectin domain or the combined C-type lectin and EGF-like domain of CD62P were expressed in bacteria and purified. How was the quality control of the purified proteins controlled (by mass spec)? The authors should include an SDS-PAGE to show the purity and size of the final products.

Response: We added SDS-PAGE of the proteins used (supplemental Fig. S1a).

3. Is it possible to increase the concentration of CD62P C-type lectin domain in their integrin binding assays (Figure 1b) to reach saturation binding and to be able to determine the K_d of binding?

Response: We increased the max. coating concentration of CD62P lectin domain to 200 $\mu\text{g/ml}$ (4x), but did not observe saturation of binding. One possibility is that the lectin domain also activates integrins probably by binding to the allosteric site (site 2) in addition to binding to the classical binding site (site 1) of integrins (the two-binding site model). If this is the case, it is not easy to determine the binding kinetics. Also, integrin-CD62P lectin interaction may be different from integrin binding to currently known integrin ligands. We will address this issue in future experiments.

4. How strong is the adhesion to the lectin domain when compared to another $\alpha 5\beta 1$ integrin ligand such as fibronectin? How long were CHO cells incubated on lectin domain ligands to assay their adhesion? The authors should show two time points (for fast <30 min and slow adhesion <45 min) and include another $\alpha 5\beta 1$ integrin ligand such as fibronectin as control.

Response: We showed time-course of adhesion with two time points (30 min and 60 min). We used 60 min for all other experiments. We were not able to perform binding experiments using biophysical methods in this revision.

We added the data that cell-surface $\alpha 4\beta 1$ and $\alpha 5\beta 1$ on CHO cells can be activated (New Fig. 6). We found that biotinylated $\alpha 4\beta 1$ or $\alpha 5\beta 1$ does not work well in ELISA-type activation assays. One possibility is that streptavidin that is used for quantification of bound integrins induce clustering of soluble integrins.

5. The activation assay (Figure 5) are interesting and important but lack controls. The authors should prepare a concentration range of CD62P C-type lectin domain (to see if and when the signal reaches saturation) and include their integrin-binding deficient CD62P C-type lectin domain mutants.

Response: To address the concern of linearity and saturation of binding of CD62P lectin binding to the integrin, we have added a dose-response curve and the effect of mutations in the lectin domain on integrin activation (new Fig. 4b,c). WT lectin domain activated integrins in a dose-dependent manner. Interestingly, the lectin mutants in the site 2-binding interface enhanced the levels of integrin activation (gain-of-function mutation), instead of inhibiting activation.

Activation was only analyzed in ELISA-type activation assays but not on cells. Is the CD62P C-type lectin domain also able to activate cell surface $\alpha\beta3$ and $\alpha1b\beta3$ integrins on living cells? What about the activation of $\alpha4\beta1$ and $\alpha5\beta1$ integrins by the CD62P C-type lectin domain?

Response: We added activation of cell-surface integrins $\alpha\beta3$, $\alpha1b\beta3$, $\alpha4\beta1$, and $\alpha5\beta1$ by the lectin domain in flow cytometry (new Fig. 5 and 6).

6. The authors state that "cRGFV or 7E3 (anti- $\alpha1b\beta3$) did not affect the binding of the lectin domain to soluble $\alpha\beta3$." However, this data is not shown. There is another instance of "data not shown" in case of inhibitors of CD62P-PSGL-1 interaction. These data should be provided, eventually as supplemental information.

Response: Supplemental Fig. S1 shows the requested results.

Minor points:

1. Figure 1D: It is written "P-selectin" in the figure but probably only the CD62P lectin domains have been used as in the other panels. The description should be changed.

Response: "P-selectin" was changed to "CD62P."

2. Unclear sentence: "The model predicts that PSGL-1 peptide in the PSGL-1-CD62P lectin domain complex bind to CD62P to integrin $\alpha\beta3$ without steric hindrance (Fig. 2a)".

Response: The sentence was divided into short sentences.

"The structure of PSGL-1 peptide (605YEYLDYDFLPETEP618) in the PSGL-1-CD62P lectin domain complex has been shown (1g1s.pdb). We generated a model of interaction of integrin, CD62P, and PSGL-1 by superposing the CD62P-integrin $\alpha\beta3$ docking model and the PSGL-1-CD62P complex. The model predicts that CD62P binds to integrin $\alpha\beta3$ without steric hindrance with PSGL-1 (Fig. 2a)."

3. Figure 2: How do the authors explain increased binding of R54K/K55E to $\alpha\beta3$ and $\alpha1b\beta3$ integrin?

Response: This could be "gain-of-function mutation".

4. Why is Figure 4 described before Figure 3 in the result section?

Response: The figures were re-organized after adding new results.

March 14, 2023

Re: Life Science Alliance manuscript #LSA-2022-01747-TR

Dr. yoshikazu takada
UC Davis
Dermatology, Biochemistry and Molecular Medicine
4645 2nd ave
4645 Second Avenue
sacramento, CA 95817

Dear Dr. Takada,

Thank you for submitting your revised manuscript entitled "The C-type lectin domain of CD62P (P-selectin) functions as an integrin ligand." to Life Science Alliance. The manuscript has been seen by the original reviewers whose comments are appended below. While the reviewers continue to be overall positive about the work in terms of its suitability for Life Science Alliance, some important issues remain.

Our general policy is that papers are considered through only one revision cycle; however, given that the suggested changes are relatively minor, we are open to one additional short round of revision. Please note that I will expect to make a final decision without additional reviewer input upon resubmission.

Please submit the final revision within one month, along with a letter that includes a point by point response to the remaining reviewer comments.

To upload the revised version of your manuscript, please log in to your account: <https://lsa.msubmit.net/cgi-bin/main.plex>
You will be guided to complete the submission of your revised manuscript and to fill in all necessary information.

B. MANUSCRIPT ORGANIZATION AND FORMATTING:

Sincerely,

Reviewer #1 (Comments to the Authors (Required)):

The revised manuscript has addressed many of the

criticisms made in my previous review but there is still a major unresolved point, related to the data in Fig 4d/e, that deserves explanation. How can it be that most mutations (termed by the authors "gain of function mutations") in the lectin domain of CD62P that are shown to reduce the binding of this domain to integrins (including the mutation R16E/K17E) do indeed enhance integrin activation? These opposing effects (reduction of integrin binding and induction of integrin activation) seem very difficult to reconcile

Reviewer #2 (Comments to the Authors (Required)):

The authors do not provide the affinity analysis suggested. Perhaps they are not familiar to Scatchard plot that is able to define 2 binding sites with different affinity. It is possible that lectin domain interacts with site 1 and 2 with different affinity. So the use of 2 ligands is not request for this approach. On the other hand the analysis of the affinity costant is requested for this type of study. I suggest the following papers: 10.1016/0003-2697(73)90195-4; 10.1002/jps.2600621233; 10.1002/jcb.240310306 ;10.1002/bip.360270106.

Reviewer #3 (Comments to the Authors (Required)):

The authors addressed most of my comments and concerns during this round of revision. There are few text/formatting issues left (see below) which can be addressed during the editing stage of the publication. I am however puzzled by one result added during the revision process. The CD62P lectin R16E/K17E mutant does not bind to Mn²⁺-activated avb3 and allbb3 integrin (Fig. 2e,f) but it strongly enhanced integrin binding to their ligands (referred to as integrin activation in the figure) (Fig. 4d,e). How can this integrin activation be specific if the CD62P lectin R16E/K17E protein does not bind active integrins? Did the authors test if the CD62P lectin R16E/K17E mutant is able to interact with integrins without prior activation Mn²⁺? How do the authors explain this finding?

I can support publication in Life Science Alliance if the authors can address this concern.

Text/formatting issues:

- Introduction - last paragraph: exchange K16E/R17E for R16E/K17E
- Supplementary Figure 1a: not referred to in the text
- I would suggest to mention that CD62P is P-selectin in the introduction and perhaps once in the results part (it is so far only written in the title). Otherwise, non-experts might be temporary lost when they read P-selectin in Figure 2.
- Supplementary Figure 1b: I am skeptical if you can draw any conclusion about the effect of the 7E3 antibody on Lectin/integrin binding if the control IgG has an even stronger effect.
- Figures: The authors should be consistent with their naming of the CD62P lectin domain in the figure legends. Currently you find "Lectin", "CD62P" or "CD62P lectin domain". In my view "CD62P lectin domain" would be most specific.

Reviewer #1 (Comments to the Authors (Required)):

The revised manuscript has addressed many of the criticisms made in my previous review but there is still a major unresolved point, related to the data in Fig 4d/e, that deserves explanation. How can it be that most mutations (termed by the authors "gain of function mutations") in the lectin domain of CD62P that are shown to reduce the binding of this domain to integrins (including the mutation R16E/K17E) do indeed enhance integrin activation? These opposing effects (reduction of integrin binding and induction of integrin activation) seem very difficult to reconcile.

Response: We do not know why R16E/K17E binds to site 2 and activates site 1. One possible explanation is that site 1 and site 2 are distinct and have different ligand-binding properties, and respond to the same ligand differently. In this sense, it is not surprising that the lectin domain mutation R16E/K17E reduces its binding to site 1 but can allosterically activate site 1 upon binding to site 2. The structural nature underlying this observation will be the subject of future studies, but we feel that it is beyond scope of the current study.

Reviewer #2 (Comments to the Authors (Required)):

The authors do not provide the affinity analysis suggested. Perhaps they are not familiar to Scatchard plot that is able to define 2 binding sites with different affinity. It is possible that lectin domain interacts with site 1 and 2 with different affinity. So the use of 2 ligands is not request for this approach. On the other hand the analysis of the affinity constant is requested for this type of study. I suggest the following papers: 10.1016/0003-2697(73)90195-4; 10.1002/jps.2600621233; 10.1002/jcb.240310306 ;10.1002/bip.360270106.

Response: The binding of soluble integrins to immobilized CD62P lectin domain in 1 mM Mn²⁺ did not saturate up to 200 µg/ml (Fig. 1b), which prevented us from measuring its K_d. Adhesion of CHO or b3-CHO cells to immobilized CD62P lectin domain in 1 mM Mg²⁺ was saturable with a K_d estimated at 15-20 µg/ml (Fig. 3a and 3b). We also examined the dose-dependence of CD62P lectin domain in integrin activation in Ca²⁺ (by binding to site 2) and found it was not saturable and thus the K_d not calculated (Fig. 4b and 4c).

Reviewer #3 (Comments to the Authors (Required)):

The authors addressed most of my comments and concerns during this round of revision. There are few text/formatting issues left (see below) which can be addressed during the editing stage of the publication. I am however puzzled by one result added during the revision process. The CD62P lectin R16E/K17E mutant does not bind to Mn²⁺-activated avb3 and aIIbb3 integrin (Fig. 2e,f) but it strongly enhanced integrin binding to their ligands (referred to as integrin activation in the figure) (Fig. 4d,e). How can this integrin activation be specific if the CD62P lectin R16E/K17E protein does not bind active integrins? Did the authors test if the CD62P lectin R16E/K17E mutant is able to interact with integrins without prior activation Mn²⁺? How do the authors explain this finding? I can support publication in Life Science Alliance if the authors can address this concern.

Response: We have edited the text throughout the manuscript and appreciate the input of the reviewer. CD62P lectin domain binds to site 1 (the classical ligand-binding site) in 1 mM Mn^{2+} and site 2 (the allosteric binding site) in 1 mM Ca^{2+} , which is distinct from site 1. It remains undetermined how R16E/K17E binds to site 2 yet allosterically activates site 1. Possible explanation is that site 1 and site 2 are distinct and have different ligand-binding properties, and allosterically respond to the same ligand differently.

How can this integrin activation be specific if the CD62P lectin R16E/K17E protein does not bind active integrins?

Response: Integrin activation was measured by binding of specific ligands (e.g., fibrinogen fragments, gC399tr or gC390-411) to site 1. We did not use the binding of CD62P lectin domain to site 1.

Text/formatting issues:

- Introduction - last paragraph: exchange K16E/R17E for R16E/K17E.

Response. Corrected

- Supplementary Figure 1a: not referred to in the text.

Response. Fig. S1 is mentioned in the method section.

- I would suggest to mention that CD62P is P-selectin in the introduction and perhaps once in the results part (it is so far only written in the title). Otherwise, non-experts might be temporarily lost when they read P-selectin in Figure 2.

Response. P-selection was mentioned in the first subtitle in the result section.

- Supplementary Figure 1b: I am skeptical if you can draw any conclusion about the effect of the 7E3 antibody on Lectin/integrin binding if the control IgG has an even stronger effect.

Response. Inhibitory antibody like 7E3 usually strongly suppresses interaction. We concluded it did not block interaction.

- Figures: The authors should be consistent with their naming of the CD62P lectin domain in the figure legends. Currently you find "Lectin", "CD62P" or "CD62P lectin domain". In my view "CD62P lectin domain" would be most specific.

Response. "CD62P lectin domain" was used throughout the manuscript and figures and legend. The manuscript appears clean.

April 11, 2023

RE: Life Science Alliance Manuscript #LSA-2022-01747-TRR

Dr. yoshikazu takada
UC Davis
Dermatology, Biochemistry and Molecular Medicine
4645 2nd ave
4645 Second Avenue
sacramento, CA 95817

Dear Dr. Takada,

Thank you for submitting your revised manuscript entitled "The C-type lectin domain of CD62P (P-selectin) functions as an integrin ligand.". We would be happy to publish your paper in Life Science Alliance pending final revisions necessary to meet our formatting guidelines.

- please add a Category for your manuscript to our system
- please use the [10 author names, et al.] format in your references (i.e. limit the author names to the first 10)

Figure Check:

*Fig S1A: after 1st column, please add a black vertical line because it's a clear splice and not continuous

A. FINAL FILES:

B. MANUSCRIPT ORGANIZATION AND FORMATTING:

Sincerely,

#1. The SDS-PAGE in Fig. S1a was not cut between markers (lane 1) and sample proteins. So we would like to publish without dark line in between.

#2. We corrected the references.

Pokharel, S. M., N. K. Shil, J. B. Gc, Z. T. Colburn, S. Y. Tsai, J. A. Segovia, T. H. Chang, S. Bandyopadhyay, S. Natesan, J. C. R. Jones et al. 2019. 'Integrin activation by the lipid molecule 25-hydroxycholesterol induces a proinflammatory response', *Nat Commun*, 10: 1482.

Wang, H. B., J. T. Wang, L. Zhang, Z. H. Geng, W. L. Xu, T. Xu, Y. Huo, X. Zhu, E. F. Plow, M. Chen et al. 2007. 'P-selectin primes leukocyte integrin activation during inflammation', *Nat Immunol*, 8: 882-92.

Weber, M. R., M. Zuka, M. Lorger, M. Tschan, B. E. Torbett, A. Zijlstra, J. P. Quigley, K. Staflin, B. P. Eliceiri, J. S. Krueger et al. 2016. 'Activated tumor cell integrin alphavbeta3 cooperates with platelets to promote extravasation and metastasis from the blood stream', *Thromb Res*, 140 Suppl 1: S27-36.

April 17, 2023

RE: Life Science Alliance Manuscript #LSA-2022-01747-TRRR

Dr. yoshikazu takada
UC Davis
Dermatology, Biochemistry and Molecular Medicine
Research III Suite 3300
4645 Second Avenue
sacramento, CA 95817

Dear Dr. takada,

Thank you for submitting your Research Article entitled "The C-type lectin domain of CD62P (P-selectin) functions as an integrin ligand.". It is a pleasure to let you know that your manuscript is now accepted for publication in Life Science Alliance. Congratulations on this interesting work.

DISTRIBUTION OF MATERIALS:

Again, congratulations on a very nice paper. I hope you found the review process to be constructive and are pleased with how the manuscript was handled editorially. We look forward to future exciting submissions from your lab.

Sincerely,
